# Minimax Optimal Online Imitation Learning via Replay Estimation

**Gokul Swamy**[*]
Carnegie Mellon University
gswamy@cmu.edu

**Nived Rajaraman**[*]
UC Berkeley
nived.rajaraman@berkeley.edu

**Matthew Peng**
UC Berkeley

**Sanjiban Choudhury**
Cornell University

**J. Andrew Bagnell**
Aurora Innovation and Carnegie Mellon University

**Zhiwei Steven Wu**
Carnegie Mellon University

**Jiantao Jiao**
UC Berkeley

**Kannan Ramchandran**
UC Berkeley

## Abstract

Online imitation learning is the problem of how best to mimic expert demonstrations, given access to the environment or an accurate simulator. Prior work has shown that in the *infinite* sample regime, exact moment matching achieves value equivalence to the expert policy. However, in the *finite* sample regime, even if one has no optimization error, empirical variance can lead to a performance gap that scales with $H^2/N_{\exp}$ for behavioral cloning and $H/\sqrt{N_{\exp}}$ for online moment matching, where $H$ is the horizon and $N_{\exp}$ is the size of the expert dataset. We introduce the technique of *replay estimation* to reduce this empirical variance: by repeatedly executing cached expert actions in a stochastic simulator, we compute a *smoother* expert visitation distribution estimate to match. In the presence of parametric function approximation, we prove a meta theorem reducing the performance gap of our approach to the *parameter estimation error* for offline classification (i.e. learning the expert policy). In the tabular setting or with linear function approximation, our meta theorem shows that the performance gap incurred by our approach achieves the optimal $\widetilde{O}\left(\min(H^{3/2}/N_{\exp}, H/\sqrt{N_{\exp}})\right)$ dependency, under significantly weaker assumptions compared to prior work. We implement multiple instantiations of our approach on several continuous control tasks and find that we are able to significantly improve policy performance across a variety of dataset sizes.

## 1 Introduction

In *online* imitation learning (IL), one is given access to *(a)* a fixed set of expert demonstrations and *(b)* an environment or simulator to perform rollouts in. Many online IL approaches fall under the umbrella of solving a *moment matching* problem between learner and expert trajectory distributions Ziebart et al. [2008], Ho and Ermon [2016],

$$\min_{\pi \in \Pi} \sup_{f \in \mathcal{F}} \mathbb{E}_{\pi}[f(s, a)] - \mathbb{E}_{\pi^E}[f(s, a)], \tag{1}$$

where $\mathbb{E}_{\pi}[\cdot]$ denotes the expectation over a random trajectory $\{(s_1, a_1), \cdots, (s_H, a_H)\}$ generated by rolling out $\pi$. Swamy et al. [2021] show that for an appropriate choice of $\mathcal{F}$, approximate solutions

---

[*]Equal contribution. Correspondence to gswamy@cmu.edu and nived.rajaraman@berkeley.edu.

36th Conference on Neural Information Processing Systems (NeurIPS 2022).

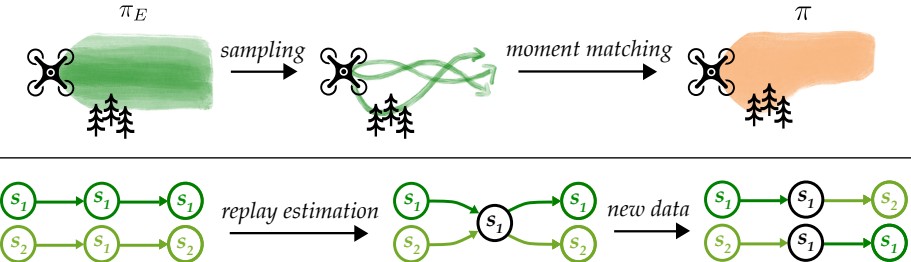

Figure 1: *Top*: Attempting to exactly match a finite-sample approximation of expert moments can cause a learner to reproduce chance occurrences (e.g. the relatively unlikely flight through the trees). This can lead to policies that perform poorly at test time (e.g. because the learner flies through the trees relatively often). *Bottom*: Replay estimation reduces the empirical variance in expert demonstrations by repeatedly executing observed expert actions in a stochastic simulator. By generating new trajectories (e.g. $s_1 \rightarrow s_1 \rightarrow s_2$ on the right) that are consistent with expert actions, one can augment the original demonstration set and compute expert moments more accurately.

of (1) have a performance gap linear in the horizon, the gold standard for sequential problems. However, a key assumption in their work is that expert moments ($\mathbb{E}_{\pi^E}[f(s,a)]$) can be estimated arbitrarily well from the available demonstrations. The resulting bounds are therefore purely a function of *optimization error*. Moving to the finite-sample regime introduces an additional concern: the *statistical error* that stems from the randomness in the data-generating process. When a small set of demonstrations are used in an adversarial optimization procedure like (1), the learner may choose to take incorrect actions in order to match noisy moments estimated from the dataset, leading to policies that perform poorly at test-time. Ideally, one would solve this problem by querying the expert for more demonstrations, as in the work of Ross et al. [2011]. However, when we are unable to do so, we still have to grapple with the question of "*how can we smooth out a noisy empirical estimate of expert moments?*"

Our answer to this question is the technique of *replay estimation*. In its most basic form, replay estimation consists of repeatedly executing observed expert actions within a stochastic simulator, terminating rollouts whenever one ventures out of the support of the expert demonstrations. Effectively, this approach stitches together parts of different trajectories to generate a smoothed estimate of expert moments. By using the simulator where we know the expert's actions, we can generate more diverse training data that is nevertheless consistent with the expert demonstrations. We argue that this technique is at once conceptually simple, practically feasible, and minimax optimal in several settings. Formally, we prove that in the worst case, behavioral cloning has a performance gap $\propto H^2/N_{\text{exp}}$, online empirical moment matching, $\propto H/\sqrt{N_{\text{exp}}}$, and our approach of replay estimation $\propto \min\{H^{3/2}/N_{\text{exp}}, H/\sqrt{N_{\text{exp}}}\}$ in the tabular setting as well as with linear function approximation. Our key insight is that *we can use a combination of simulated and empirical rollouts to optimally estimate expert moments.* We can then plug this improved estimate into a variety of moment-matching algorithms, for strong test-time policy performance. More explicitly, our work makes the following three contributions:

1. We extend replay estimation (RE) Rajaraman et al. [2020] beyond the tabular and deterministic setting by introducing the notion of a *soft membership oracle* and *prefix weights*.

2. We show how to instantiate the membership oracles for IL with parametric function approximation and prove a meta-theorem relating the imitation gap of RE to the parameter estimation error for offline classification on the dataset (Theorem 3). Instantiating our main result in the case of linear function approximation, we show how to achieve the best known imitation gap of $\widetilde{O}\left(H^{3/2}d^{5/4}/N_{\text{exp}}\right)$ under significantly weaker assumptions compared to prior work Rajaraman et al. [2021].

3. We give multiple practical options for constructing performant membership oracles. We then use these approximate oracles to significantly improve the performance of online IL on several continuous control tasks across a variety of dataset sizes. We also investigate the differences between our proposed oracles.

We sketch the benefits of replay estimation before providing theoretical and empirical evidence to support our claims.

## 2 The Replay Estimator

We begin with a tabular vignette to illustrate our key insight in greater detail. We compare two algorithms: offline behavioral cloning (BC) Pomerleau [1989] and online moment matching (MM) Swamy et al. [2021]. Throughout, we focus on learning policies from finite samples.

**Suboptimality of Empirical Moment Matching.** Consider the MDP in Fig. 2, where the expert always takes the green action. Doing so puts them in $s_1$ or $s_2$ with equal probability. Given that the expert is deterministic and there are few states, BC could easily recover the expert's policy by learning to simply output the observed green action on both states, even when there are very few demonstrations.

Now, what would happen if we tried to match moments of the expert's state-action visitation distribution for this problem? It is rather unlikely that we see *exactly* equal probabilities for both states in the observed data. If by chance we see $s_2$ more than we see $s_1$, the learner might realize that the only way to match the observed state distribution (a prerequisite for matching the observed state-action distribution) is to occasionally take the red action at $s_2$. In general, this could cause the learner to spend an unnecessary amount of time in $s_2$ which may be undesirable (e.g. if $s_2$ corresponds to the tree-filled area in Fig. 1 (top)). The core issue we hope to illustrate in this example is that by treating the empirical estimate of the expert's behavior as perfectly accurate, distribution matching can force the learner to take incorrect actions to minimize training error, leading to test-time performance degradation. As we will discuss in Sec. 3, this can lead to slow statistical rates $\propto H/\sqrt{N_{\exp}}$.

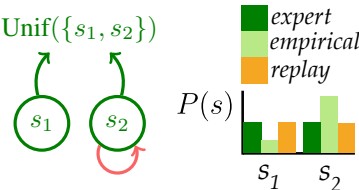

Figure 2: An MDP where the expert always takes the green action that puts them in the uniform distribution over $s_1$ and $s_2$. Because of full expert support, BC will learn to always take this action at both states. However, if the empirical state distribution is more tilted towards $s_2$, MM will take the incorrect red action.

**Suboptimality of Behavioral Cloning.** Because it does not account for the covariate shift that results from policy action choices, behavioral cloning can lead to a quadratic compounding of errors and poor test time performance Ross et al. [2011]. Consider, for example, the MDP in Fig. 3.

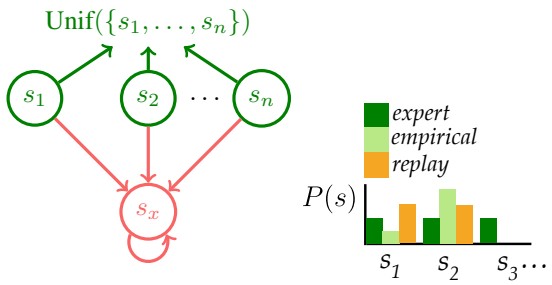

Figure 3: The expert always takes the green action, which places it in a uniform distribution over $s_1, \ldots, s_n$. At states where we have demonstrations (e.g. $s_1$, $s_2$), both BC and MM will take the same, correct action. However, at states where we have no demonstrations (e.g. $s_3$), MM will correctly take the green action to get back to states with demonstration support, while BC might not.

Let us assume that the expert always takes the green action, dropping them in a state in the top row with uniform probability. In a small demonstration set, we might not see expert actions at some states in the top row. At all such states, BC will have no idea of what to do. In contrast, MM will take the green action as doing so might send the learner back to a state with positive demonstration support. Thus for this problem, MM will recover the optimal policy while BC will not. As we will discuss in Sec. 3, this leads to errors $\propto H^2/N_{\exp}$ in the worst case.

**Replay Estimation.** The previous two examples show us that there exist simple MDPs for which BC or MM will not recover the expert's policy. This begs the question: is it possible to do

*better than both worlds* and recover the optimal policy on both problems with a single algorithm? It turns out it is indeed possible to do so, via the technique of *replay estimation*. In its simplest form,

replay estimation involves playing a cached expert action whenever possible and re-starting the rollout if one ventures out of the support of the demonstrations. Then, one appends these rollouts to the demonstration set, treating them as additional training data – while biased, they are consistent with observed expert behavior. Intuitively, repeated simulation has a *smoothing* effect on the training data as doing so marginalizes out the statistical error that comes from the stochasticity of the dynamics. We can see this point more explicitly by considering the above two MDP examples: in Fig. 2, repeatedly playing the green action and appending these rollouts to the expert dataset would bring us much closer to a uniform distribution over $s_1$ and $s_2$. Similarly, in Fig. 3, replay estimation would bring us toward a uniform distribution over the states $\{s_1, \cdots, s_n\}$ in the expert demonstrations.

We could then plug in this improved distribution estimate into the MM procedure (1). Notice how doing so would cause MM to be highly likely to recover the optimal policy on both MDPs. For example, in Fig. 2, replay estimation would make the learner much less likely to play the red action in $s_2$, It turns out this fact is sufficient to establish *statistical optimality* in the tabular setting and with linear function approximation, with an error rate $\propto \min(H^{3/2}/N_{\exp}, H/\sqrt{N_{\exp}})$. In short, replay estimation is a practical technique for reducing some of the finite-sample variance in expert demonstrations that enables MM to perform optimally in the finite sample regime. We now provide some intuition on how to generalize this approach to beyond the tabular setting.

**Leaving the Tabular Setting.** The prior work of Rajaraman et al. [2020] considers the tabular setting; this characteristic makes it easy to answer the question of "*on what states do we know the expert's action?*" To enable us to answer this question more generally, we introduce the notion of a *membership oracle* $\mathcal{M} : \mathcal{S} \to \{0, 1\}$. Explicitly, $\mathcal{M}(s) = 1$ for states where we know the expert's action well (e.g. states where we have lots of similar demonstrations) and $\mathcal{M}(s) = 0$ otherwise. Alternatively, $\mathcal{M}(s) = 1$ on states where BC , which attempts to directly output expert actions, is accurate and 0 otherwise.

We can then compute expert moments by splitting on the output of the membership oracle:

$$\mathbb{E}_{\pi^E}[f(s,a)] = \underbrace{\mathbb{E}_{\pi^E}\left[f(s,a)\mathbb{1}(\mathcal{M}(s) = 1)\right]}_{(i)} + \underbrace{\mathbb{E}_{\pi^E}\left[f(s,a)\mathbb{1}(\mathcal{M}(s) = 0)\right]}_{(ii)} \tag{2}$$

Note that the indicators in $(i)$ and $(ii)$ are complements of each other, rendering the above sum a valid estimate of the expert moment. As we know the expert action well wherever $\mathcal{M}(s) = 1$, simulated rollouts of the BC policy approximates $(i)$ well; on the other hand we resort to a naive empirical estimate to approximate $(ii)$, as we do not know enough about the expert's action at these states to accurately generate additional demonstrations via BC rollouts. In general, we relax $\mathcal{M}$ to a *soft membership oracle* Zadeh [1965], in order to handle uncertainty in how well we know the expert's action at a given state. We proceed by first analyzing the statistical properties of applying MM to this bipartite estimator before discussing practical constructions of performant membership oracles.

## 3 Theoretical Analysis

The proofs of all results in this section are deferred to appendix A. We begin by introducing some notation.

**Notation.** Let $\Delta(X)$ denote the probability simplex over set $X$ and let $\gtrsim, \lesssim, \asymp$ respectively denote greater than, lesser than and equality up to constants. We study the IL problem in the episodic MDP setting with state space $\mathcal{S}$, action space $\mathcal{A}$ and horizon $H$. We assume that the transition, reward function and policies can be non-stationary. The MDP transition is denoted $P = \{\rho, P_1, \cdots, P_{H-1}\}$, where $\rho$ is the initial state distribution and $P_t : \mathcal{S} \times \mathcal{A} \to \Delta(\mathcal{S})$, while the reward function is denoted $r = \{r_1, \cdots, r_H\}$ where $r_t : \mathcal{S} \times \mathcal{A} \to [0, 1]$. In the online imitation learning setting, the learner has access to a finite dataset $D$ of $N_{\exp}$ trajectories (i.e. the sequences of states visited and actions played) generated by rolling out expert policy $\pi^E$. Importantly, the learner *does not observe rewards during rollouts*. The fundamental goal of the learner is to learn a policy $\widehat{\pi}$ such that the

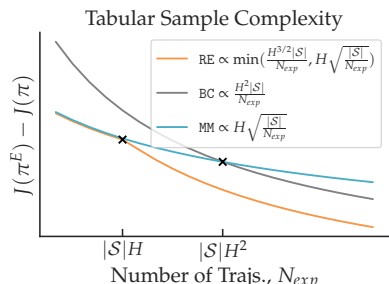

Figure 4: In the tabular setting, RE inherits the superior low-data performance of moment-matching approaches and is able to perform better than both MM and BC with enough data.

*imitation gap*, $J(\pi^E) - J(\widehat{\pi})$ is small. Here, $J(\pi)$ denotes
the expected value of the policy $\pi$, $\mathbb{E}_\pi \left[ \sum_{t=1}^H r_t(s_t, a_t) \right]$. $\mathbb{E}_D[\cdot]$ denotes the empirical expectation computed using trajectories from the dataset $D$.

**Behavior cloning.** A standard approach for imitation learning is behavioral cloning (BC) Pomerleau [1989], which trains a classifier from expert states to expert actions. More formally, for the 0-1 loss $\ell$ (or a continuous proxy) BC minimizes the empirical classification error,

$$\pi^{\texttt{BC}} \leftarrow \arg\min_{\pi \in \Pi} \mathbb{E}_D \left[ \frac{1}{H} \sum_{t=1}^H \ell(\pi_t(s_t), a_t) \right]. \tag{3}$$

It is known from Ross et al. [2011] and Rajaraman et al. [2020] that in the tabular setting, the expected imitation gap for BC is always $\lesssim |\mathcal{S}|H^2/N_{\exp}$. The fast $1/N_{\exp}$ rate comes from the fact that on states where the learner observes the deterministic expert's actions, BC simply replays them. Thus, the performance gap is proportional to the mass of unseen states, which decays as $1/N_{\exp}$. The $H^2$ dependence comes from the fact that a single mistake can take the learner out of the distribution of the expert, causing it to make mistakes for the rest of the horizon. Importantly, this bound is tight – i.e. there exists an MDP instance on which BC incurs this imitation gap (See Appendix A.3).

**Theorem 1** (Theorem 6.1 of Rajaraman et al. [2020])**.** *Then, there exists a tabular MDP instance such that the BC incurs,* $\mathbb{E}\left[ J(\pi^E) - J(\pi^{\texttt{BC}}) \right] \gtrsim \min\left\{ H, |\mathcal{S}|H^2/N_{\exp} \right\}.$

**Moment matching.** Many standard algorithms in the IL literature fall under the empirical moment matching framework (e.g. GAIL Ho and Ermon [2016], MaxEnt IRL Ziebart et al. [2008]); see Table 3 of Swamy et al. [2021] for more examples. Reward moment matching corresponds to finding a policy which best matches the state-action visitation measure of $\pi^E$, in the sense of minimizing an Integral Probability Metric (IPM) Müller [1997]. In the finite sample setting, the *empirical moment matching* learner $\pi^{\texttt{MM}}$ attempts to best match the empirical state-visitation measure. Namely,

$$\pi^{\texttt{MM}} \in \arg\min_{\pi \in \Pi} \sup_{f \in \mathcal{F}} \mathbb{E}_\pi \left[ \frac{\sum_{t=1}^H f_t(s_t, a_t)}{H} \right] - \mathbb{E}_D \left[ \frac{\sum_{t=1}^H f_t(s_t, a_t)}{H} \right]. \tag{4}$$

First we show an upper bound on the imitation gap incurred by empirical moment matching.

**Theorem 3.1.** *Consider the empirical moment matching learner $\pi^{\texttt{MM}}$ (eq. (4)), instantiated with an appropriate discriminator class $\mathcal{F}$. The imitation gap satisfies* $\mathbb{E}\left[ J(\pi^E) - J(\pi^{\texttt{MM}}) \right] \lesssim H\sqrt{|\mathcal{S}|/N_{\exp}}.$

The proof of this result can be found in Appendix A.2. It turns out that this guarantee is essentially tight for empirical moment matching, answering an open question from Rajaraman et al. [2020].

**Theorem 3.2.** *If $H \geq 4$, there is a tabular IL instance with 2 states and actions on which with constant probability, the empirical moment matching learner (eq. (4)) incurs,* $J(\pi^E) - J(\pi^{\texttt{MM}}) \gtrsim H/\sqrt{N_{\exp}}.$

The proof of this result is deferred to appendix A.4. The proof of this lower bound exploits the fact that the data generation process in the dataset is inherently random. Consider a slight modification of the MDP instance shown in fig. 2, where the reward function is 0 for $t = 1$. For $t \geq 2$, the transition function is absorbing at both states; the reward function equals 1 at the state $s_1$ for any action and is 0 everywhere else. Then, the expert state distribution at time 2 and every time thereon is in uniform across the two states, $\{1/2, 1/2\}$. However, in the dataset $D$, the learner sees a noisy realization of this distribution in the dataset of the form $\{1/2 - \delta, 1/2 + \delta\}$ for $|\delta| \approx \pm 1/\sqrt{N_{\exp}}$. Because of this noise, the empirical moment matching learner may be encouraged to *deviate from the expert's observed behavior* and pick the red action at $s_2$ as this results in a better match to the empirical *state* visitation measures at every point in the rest of the episode - a prerequisite to matching the empirical state-action visitation measure. The learner is willing to pick an action different from what the expert played in order to better match the *inherently noisy* empirical state-action visitation distribution.

**Remark 1.** *Theorems 1 and 3.2 are separate lower bound IL instances against the performance of BC and empirical moment matching. On the uniform mixture of the two MDPs (i.e. deciding the underlying MDP based on the outcome of a fair coin), with constant probability, both $J(\pi^E) - J(\pi^{\texttt{BC}}) \gtrsim |\mathcal{S}|H^2/N_{\exp}$ and $J(\pi^E) - J(\pi^{\texttt{MM}}) \gtrsim H/\sqrt{N_{\exp}}$. On this mixture instance, training both BC and empirical moment matching and choosing the better of the two is also statistically suboptimal.*

## 3.1 Replay Estimation

A natural question at this point is whether there is an algorithm that is *better than both worlds*, i.e. algorithm which can outperform the worst case imitation gap of empirical moment matching and BC . The answer to this question in the tabular setting was recently provided by Rajaraman et al. [2020] who propose `Mimic-MD` , achieving better performance than both BC and MM . This improvement is possible because BC does not use any dynamics information and MM does not leverage the knowledge of where expert actions are known. However, it is unclear how to extend this approach beyond the tabular setting as the algorithm relies on a large measure of states being visited in the demonstrations.

---

**Algorithm 1:** Replay Estimation (RE )

**Input:** Expert demonstrations $D$, policy class $\Pi$, moment class $\mathcal{F} = \bigoplus_{t=1}^{H} \mathcal{F}_t$, simulator SIM, ALG which returns a membership oracle given a dataset;

1 Partition the dataset $D$ into $D_1$ and $D_2$;
2 Using ALG, train the membership oracle $\mathcal{M}$ on $D_1$;
3 Train $\pi^{\text{BC}}$ using behavior cloning on $D_1$;
4 Roll out $\pi^{\text{BC}}$ in SIM $N_{\text{replay}}$ times to construct a new dataset, $D_{\text{replay}}$;
5 Define prefix weights $\mathcal{P}(s_{1\ldots t}) = \prod_{t'=1}^{t} \mathcal{M}(s_{t'}, t')$;
6 Define,

$$
7 \quad \widehat{E}(f) = \mathbb{E}_{D_{\text{replay}}} \left[ \frac{1}{H} \sum_{t=1}^{H} f_t(s_t, a_t) \left( \mathcal{P}(s_{1\ldots t}) \right) \right] + \mathbb{E}_{D_2} \left[ \frac{1}{H} \sum_{t=1}^{H} f_t(s_t, a_t) \left( 1 - \mathcal{P}(s_{1\ldots t}) \right) \right].
$$

**Output:** $\pi^{\text{RE}}$, a solution to the moment-matching problem:

$$
\arg\min_{\pi \in \Pi} \sup_{f \in \mathcal{F}} \mathbb{E}_\pi \left[ \frac{1}{H} \sum_{t=1}^{H} f_t(s_t, a_t) \right] - \widehat{E}(f) \tag{5}
$$

---

To handle this challenge, we introduce the notion of a *soft membership oracle* Zadeh [1965], $\mathcal{M} : \mathcal{S} \times [H] \to [0, 1]$ which captures the learner's inherent uncertainty in the expert's actions at a state at each point in an episode. The soft membership oracle assigns high weight to a state if BC is likely to closely agree with the expert policy and gives a lower weight to states where BC is likely to be inaccurate. By this definition, if the membership oracle is consistently large at all the states visited in a trajectory, we can be confident that *a trajectory generated by BC is as though it was a rollout from the expert policy*. Formally, for any function $g$ and time $t = 1, \cdots, H$, we have the decomposition,

$$
\mathbb{E}_{\pi^E}[g(s_t, a_t)] = \underbrace{\mathbb{E}_{\pi^E}[g(s_t, a_t)\mathcal{P}(s_{1\ldots t})]}_{(i)} + \underbrace{\mathbb{E}_{\pi^E}[g(s_t, a_t)(1 - \mathcal{P}(s_{1\ldots t}))]}_{(ii)} \tag{6}
$$

where $\mathcal{P}(s_{1\ldots t})$ is defined as the *prefix weight* $\prod_{t'=1}^{t} \mathcal{M}(s_{t'}, t')$. We need to use prefix weights instead of the single-sample weights sketched in the previous section to account for the probability of BC getting to the current state in the same manner the expert would have. Because of the high accuracy of BC on segments with high prefix weights, in eq. (6), $(i)$ can be approximated by replacing the expectation over $\pi^E$ by that over $\pi^{\text{BC}}$, i.e. replay estimation. On the other hand, since the prefix weight is low on the remaining trajectories in $(ii)$, we know that BC is inaccurate, so we resort to using a simple empirical estimate to estimate this term.

While we leave the particular choice of the soft membership oracle flexible, intuitively, states at which BC closely agrees with the expert policy should be given high weight while where those where BC is inaccurate should be weighted lower. In Section 4, we discuss several practical approaches to designing such a soft membership oracle. We first prove a generic policy performance guarantee for the outputs of our algorithm as a function of the choice of $\mathcal{M}$.

**Theorem 2.** *Consider the policy $\pi^{RE}$ returned by Algorithm 1. Assume that $\pi^E \in \Pi$ and the ground truth reward function $r_t \in \mathcal{F}_t$, which is assumed to be symmetric ($f_t \in \mathcal{F}_t \iff -f_t \in \mathcal{F}_t$) and bounded (For all $f_t \in \mathcal{F}_t$, $\|f_t\|_\infty \leq 1$). Choose $|D_1|, |D_2| = \Theta(N_{\text{exp}})$ and suppose $N_{\text{replay}} \to \infty$.*

*With probability* $\geq 1 - 3\delta$,

$$J(\pi^E) - J(\pi^{RE}) \lesssim \mathcal{L}_1 + \mathcal{L}_2 + \frac{\log\left(F_{\max}H/\delta\right)}{N_{\exp}} \tag{7}$$

*where* $F_{\max} \triangleq \max_{t\in[H]} |\mathcal{F}_t|$, *and,*

$$\mathcal{L}_1 \triangleq H^2 \, \mathbb{E}_{\pi^E}\left[\frac{\sum_{t=1}^H \mathcal{M}(s_t, t)\mathsf{TV}\left(\pi_t^E(\cdot|s_t), \pi_t^{BC}(\cdot|s_t)\right)}{H}\right], \tag{8}$$

$$\mathcal{L}_2 \triangleq H^{3/2}\sqrt{\frac{\log\left(F_{\max}H/\delta\right)}{N_{\exp}}\frac{\sum_{t=1}^H \mathbb{E}_{\pi^E}\left[1 - \mathcal{M}(s_t, t)\right]}{H}}.$$

We discuss a proof of this result in Appendix B and include bounds when $N_{\mathrm{replay}}$ is finite.

**Remark 2.** *Note that Theorem 2 can be extended to infinite function families using the standard technique of replacing* $|\mathcal{F}_t|$ *by the* $\epsilon$ *log-covering number of* $\mathcal{F}_t$ *in say, the* $L_2$ *norm, for* $\epsilon = \frac{1}{N_{\exp}}$. *For ease of exposition here, we stick to the case where* $\mathcal{F}_t$ *is finite.*

The term $\mathcal{L}_1$ measures how accurate BC is on states from expert trajectories where $\mathcal{M}(s_t, t)$ is large. Intuitively, if we set $\mathcal{M}(s_t, t) = 1$ on states where BC is accurate and $\mathcal{M}(s_t, t) = 0$ elsewhere, we would expect this term to be small. $\mathcal{L}_2$ can be thought of a measure of BC 's coverage: it tells us how much of the expert's visitation distribution we believe BC to be inaccurate on. If BC has good coverage (i.e. $1 - \mathcal{M}(s_t, t)$ is small on expert trajectories), we expect this term to be small.

Prima facie, one might think that because $\mathcal{L}_1$ resembles the imitation gap of BC and $\mathcal{L}_2$ resembles that of MM , RE can only perform as well as the best of BC ($\propto H^2/N_{\exp}$) and MM ($\propto H/\sqrt{N_{\exp}}$) on a given instance. However, with a careful choice of $\mathcal{M}$, one can achieve "better than both worlds" statistical rates. In particular, since RE is a generalization of Mimic-MD of Rajaraman et al. [2020], in the tabular setting, an appropriately initialized version of RE achieves the optimal imitation gap of $\min\left\{\frac{|\mathcal{S}|H^{3/2}}{N_{\exp}}, H\sqrt{\frac{|\mathcal{S}|}{N_{\exp}}}\right\}\log\left(\frac{|\mathcal{S}|H}{\delta}\right)$ and strictly improves over both BC and MM .

We now show how to extend this result and instantiate the membership oracle for parametric function approximation and provide a statistical guarantee under a particular margin assumption.

## 3.2 Parametric Function Approximation: Reduction to Offline Classification

While BC can be thought of as a reduction of IL to offline classification, the algorithm does not take into account the knowledge of the transition of the MDP. This is reflected in the quadratic dependency in the horizon, error compounding Ross et al. [2011]. In this section, we study a novel reduction of IL with parametric function approximation to *parameter estimation in offline classification*, which we define formally. We provide a provable guarantee, assuming the learner has access to a classification oracle and the underlying function class admits a Lipschitz parameterization.

**Definition 1** (IL with function-approximation). *In this setting, for each* $t \in [H]$, *there is a parameter class* $\Theta_t \subseteq \mathbb{B}_2^d$, *the unit* $L_2$ *ball in* $d$ *dimensions, and an associated function class* $\{f_{\theta_t} : \theta_t \in \Theta_t\}$. *For each* $t \in [H]$ *there exists an unknown* $\theta_t^E \in \Theta_t$ *such that* $\forall s \in \mathcal{S}$,

$$\pi_t^E(s) = \arg\max_{a\in\mathcal{A}} f_{\theta_t^E}(s, a). \tag{9}$$

**Definition 2** (Lipschitz parameterization). *A function class* $\mathcal{G} = \{g_\theta : \theta \in \Theta\}$ *where* $g_\theta(\cdot) : \mathcal{X} \to \mathbb{R}$ *is said to satisfy L-Lipschitz parameterization if,* $\|g_\theta(\cdot) - g_{\theta'}(\cdot)\|_\infty \leq L\|\theta - \theta'\|_2$. *In other words, for each* $x \in \mathcal{X}$, $g_\theta(x)$ *is an L-Lipschitz function in* $\theta$, *in the* $L_2$ *norm.*

**Assumption 1.** *For each* $t$, *the class* $\{f_{\theta_t} : \theta_t \in \Theta_t\}$ *is L-Lipschitz in its parameterization,* $\theta_t \in \Theta_t$.

To deal with parametric function approximation, we assume that the learner has access to an *offline classification oracle* which, given a dataset of classification examples, approximately returns the underlying ground truth parameter. More formally,

**Assumption 2** (Offline classification oracle). *We assume that the learner has access to a multi-class classification oracle, which given $n$ examples of the form, $(s^i, a^i)$ where $s^i \overset{i.i.d.}{\sim} \mathcal{D}$ and $a^i = \arg\max_{a \in \mathcal{A}} f_{\theta^*}(s^i, a)$, returns a $\hat{\theta} \in \Theta$ such that, with probability $\geq 1 - \delta$, $\|\hat{\theta} - \theta^*\|_2 \leq \mathcal{E}_{\Theta, n, \delta}$.*

We assume that this classification oracle is used by RE to train the BC policy in Line 3 of Algorithm 1.

A careful reader might note that Assumption 2 asks for a slightly stronger requirement than just finding a classifier with small generalization error (which need not be close to the ground truth $\theta^*$). The latter problem was studied in Daniely et al. [2013] who show that the *Natarajan dimension*, up to log-factors in the number of classes (i.e. number of actions) captures the generalization error of the best learner, which scales as $\Theta(1/n)$ given $n$ classification examples. Under some mild regularity (coverage) assumptions on the input distribution $\mathcal{D}$, we show that the generalization guarantee carries over to learning the parameter $\theta$, for example in linear classification (Appendix B.4). In particular, we show that $\mathcal{E}_{\Theta, n, \delta} \lesssim \frac{d + \log(1/\delta)}{n}$ when $f_\theta = \langle \theta, \cdot \rangle$. The membership oracle we study is defined below,

$$\mathcal{M}(s, t) = \begin{cases} +1 & \text{if } \exists a \in \mathcal{A} \text{ such that, } \forall a' \in \mathcal{A}, \ f_{\hat{\theta}_t^{\text{BC}}}(s, a) - f_{\hat{\theta}_t^{\text{BC}}}(s, a') \geq 2L\mathcal{E}_{\Theta_t, N_{\text{exp}}, \delta/H} \\ 0 & \text{otherwise.} \end{cases} \tag{10}$$

Intuitively, $\mathcal{M}$ assigns a state as $+1$ if BC classifies it with a significant margin as some action.

Finally, we introduce the main assumption on the IL instances we study. We assume that the classification problems solved by BC at each $t \in [H]$ satisfy a margin condition.

**Assumption 3** (Weak margin condition). *The weak margin condition assumes that for each $t$, there is no classifier $\theta \in \Theta_t$ such that for a large mass of states, $f_\theta(s_t, \pi_t^E(s_t)) - \max_{a \neq \pi_t^E(s_t)} f_\theta(s_t, a)$, i.e. the "margin" from the nearest classification boundary, is small. Formally, the weak-margin condition with parameter $\mu$ states that, for any $\theta \in \Theta_t$ and $\eta \leq 1/\mu$,*

$$\Pr_{\pi^E} \left( f_\theta(s_t, \pi_t^E(s)) - \max_{a \neq \pi_t^E(s_t)} f_\theta(s_t, a) \geq \eta \right) \geq e^{-\mu\eta}. \tag{11}$$

*The weak margin condition only assumes that there is at least an exponentially small (in $\eta$) mass of states with margin at least $\eta$. Smaller $\mu$ indicates a larger mass away from any decision boundary. It suffices to assume that eq. (11) is only true for $\theta$ as the classifier in Assumption 2 for our guarantees (Theorem 3) to hold.*

Under these three assumptions and using the membership oracle defined above, we can provide a strong guarantee for RE :

**Theorem 3.** *For IL with parametric function approximation, under Assumptions 1 to 3, appropriately instatiating RE ensures that with probability $\geq 1 - 4\delta$,*

$$J(\pi^E) - J(\pi^{RE}) \lesssim H^{3/2} \sqrt{\frac{\mu L \log(F_{\max}H/\delta)}{N_{\text{exp}}} \frac{\sum_{t=1}^{H} \mathcal{E}_{\Theta_t, N_{\text{exp}}, \delta/H}}{H}} + \frac{\log(F_{\max}H/\delta)}{N_{\text{exp}}}. \tag{12}$$

In Appendix B.4, we explicitly instantiate these guarantees for the special case of where the expert follows a linear classifier acting on a known set of feature representations of the state-action pairs. The prior work of Rajaraman et al. [2021] shows an imitation gap of $\widetilde{O}\left(H^{3/2} d^{5/4}/N_{\text{exp}}\right)$ for RE in the linear expert setting, under the restrictive assumptions of a binary action space, and a certain uniform distribution assumption on the feature distribution. In contrast, the guarantee of Theorem 3 can be used to achieve the same guarantee, for general action spaces and under a significant weakening of the uniform feature distribution constraint to a multi-class analog of the *strong distribution assumption* Audibert and Tsybakov [2007]. Under this assumption, we show that the best achievable parameter recovery error, $\mathcal{E}_{\Theta_t, N_{\text{exp}}, \delta/H}$ and the best achievable 0-1 generalization error ($\mathbb{E}_{s \sim \mathcal{D}}[\max_{a \in \mathcal{A}} f_{\hat{\theta}}(s, a) \neq \max_{a \in \mathcal{A}} f_{\theta^*}(s, a)]$, in the notation of Assumption 2) for classification, $\mathcal{E}_{\Theta_t, N_{\text{exp}}, \delta/H}^{\text{class}}$, *match up to problem dependent constants.*

The best known statistical guarantees on imitation gap for BC (Ross et al. [2011]) and MM are,

$$J(\pi^E) - J(\pi^{BC}) \leq \text{Gap}(\pi^{BC}) \triangleq H^2 \frac{\sum_{t=1}^{H} \mathcal{E}_{\Theta_t, N_{\text{exp}}, \delta/H}^{\text{class}}}{H}, \text{ and} \tag{13}$$

$$J(\pi^E) - J(\pi^{MM}) \leq \text{Gap}(\pi^{MM}) \triangleq H \sqrt{\frac{\log(F_{\max}H/\delta)}{N_{\text{exp}}}} \tag{14}$$

The guarantee in Theorem 3 for RE therefore grows as $\mathrm{Gap}(\pi^{\mathrm{MM}})\sqrt{\mathrm{Gap}(\pi^{\mathrm{BC}})/H}$, whenever we are able to establish that $\mathcal{E}_{\Theta_t,N_{\mathrm{exp}},\delta/H} \asymp \mathcal{E}^{\mathrm{class}}_{\Theta_t,N_{\mathrm{exp}},\delta/H}$ up to problem dependent constants. Our guarantees for RE in the presence of parametric function approximation thus give us reasons to expect RE to improve the performance of MM . We now turn our attention to validating this in practice.

## 4 Practical Algorithm

When considering RE (Alg. 1), two main questions arise: *(i)* How does one construct a membership oracle in practice, especially when action spaces may be continuous?, and *(ii)* How does one solve the moment matching problem (eq. (5))? We now provide potential answers to both of these questions.

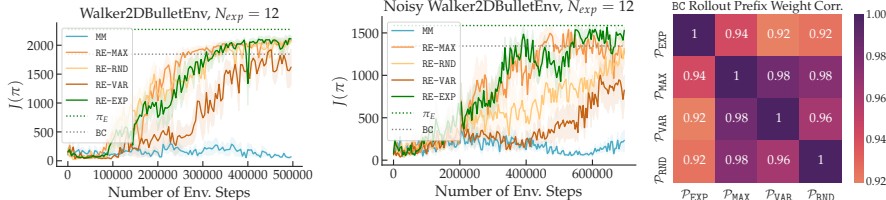

Figure 5: **Left:** All variants of RE are able to nearly match expert performance while MM struggles to make any progress. **Center:** We add i.i.d. noise to the environment to make the control problem more challenging. RE is still able to match expert performance, unlike MM . **Right:** We compute correlations between the idealized prefix weights of $\mathcal{M}_{\mathrm{EXP}}$ and the other oracles and see $\mathcal{M}_{\mathrm{MAX}}$ correlate most.

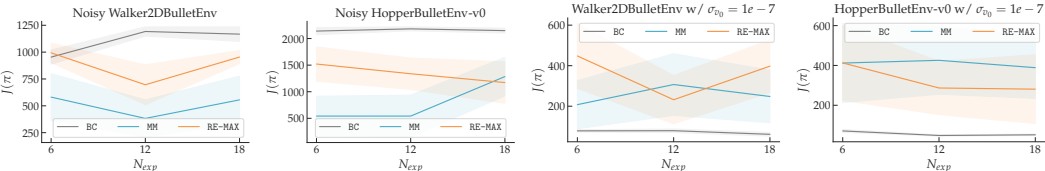

Figure 6: We see RE with $\mathcal{M}_{\mathrm{MAX}}$ improve the performance of MM on the Noisy Walker2DBulletEnv and HopperBulletEnv tasks. We see RE (and MM ) out-perform BC on the initial-state-perturbed Walker2DBulletEnv and HopperBulletEnv tasks .

**Membership Oracle.** $\mathcal{M}$ represents how uncertain BC is about the expert's action. Ideally, we would be able to query the expert policy for an action and weight by the distance between the BC and expert actions, e.g. for sigmoid function $\sigma$ and constants $\mu, \beta$,

$$\mathcal{M}_{\mathrm{EXP}}(s,t) = \sigma\left(\frac{\mu - \left\|\pi^{\mathrm{BC}}(s) - \pi^E(s)\right\|_2}{\beta}\right), \tag{15}$$

However, in the non-interactive setting, we can only approximate this quantity. The first approximation we propose using is inspired by Random Network Distillation (RND) [Burda et al., 2018], used by Wang et al. [2019] to estimate the support of the expert policy. We instead propose using RND as a measure of how uncertain BC is about expert actions. That is,

$$\mathcal{M}_{\mathrm{RND}}(s,t) = \sigma\left(\frac{\mu - \left\|\pi^{\mathrm{BC}}(s) - \widehat{\pi^{\mathrm{BC}}}(s)\right\|_2}{\beta}\right), \tag{16}$$

where $\widehat{\pi^{\mathrm{BC}}}$ is a network trained to imitate $\pi^{\mathrm{BC}}$ on expert states. To train $\widehat{\pi^{\mathrm{BC}}}$, we evaluate $\pi^{\mathrm{BC}}$ on expert states and plug this new dataset into their standard IL pipeline.

We can also utilize other uncertainty measures like the disagreement of an ensemble, which has shown success on various simulated sequential decision making tasks [Pathak et al., 2019]. Past work by Brantley et al. [2019] utilizes the variance of a set of BC learners as a regularizer on top of standard

BC error. We instead propose using it to *weight* the two halves of our estimator. Instead, we define,

$$\mathcal{M}_{\text{VAR}}(s,t) = \sigma\left(\frac{\mu - \text{Var}(\pi^{\text{BC}(1)}(s), \ldots, \pi^{\text{BC}(k)}(s))}{\beta}\right), \tag{17}$$

where $\{\pi^{\text{BC}(1)}, \cdots, \pi^{\text{BC}(k)}\}$ are BC policies trained with different initializations, which produces sufficient diversity when using deep networks as function approximators. Lastly, we can also use the maximum difference in the ensemble Kidambi et al. [2021] as our disagreement measure:

$$\mathcal{M}_{\text{MAX}}(s,t) = \sigma\left(\frac{\mu - \max_{i,j\in[k]}\left\|\pi_i^{BC}(s) - \pi_j^{BC}(s)\right\|_2}{\beta}\right), \tag{18}$$

For computing prefix weights, we use the average of distances up till the current timestep instead of the sum of distances that would follow directly from the idealized algorithm presented in Sec. 3. This modification serves to improve the numerical stability of our method.

**Moment Matching.** We implement approximate Nash equilibrium computation of (5) by running a no-regret learner against a best-response counterpart [Swamy et al., 2021]. Our approach is related to the GAIL algorithm of Ho and Ermon [2016] which we improve in 4 ways: *(i)* we use a general Integral Probability Metric Müller [1997] instead of the Jensen-Shannon Divergence used in the original paper which improves the representation power of the the discriminator, *(ii)* we add in gradient penalties to the discriminator, which improves convergence rates Gulrajani et al. [2017], *(iii)* we solve the entropy-regularized forward problem via Soft-Actor Critic Haarnoja et al. [2018] as the policy optimizer, for improved sample efficiency, and *(iv)* we use optimistic mirror descent instead of gradient descent as our optimization algorithm for both players, giving us faster convergence to Nash equilibria, both in theory Syrgkanis et al. [2015] and in practice Daskalakis et al. [2017]. Together, these changes lead to an implementation which *significantly out-performs the original*, giving us a strong baseline to compare against. We include an ablation to confirm this fact in Appendix C. We emphasize that the RE technique can be used to improve *any* online moment matching algorithm and that the above description is merely the approach we chose for this paper.

## 5   Experimental Results

We now quantify the empirical benefits of RE on several continuous control tasks from the the PyBullet suite Coumans and Bai [2016–2019]. All the task we consider have long horizons ($H \approx 1000$) and we use relatively few demonstrations. ($N_{\text{exp}} \leq 20$). We set $N_{\text{replay}}$ as 100 BC rollouts (Line 4 of Algorithm 1). We test all four membership oracles from the previous section ($\mathcal{M}_{\text{EXP}}$ as an idealized target, $\mathcal{M}_{\text{RND}}$, $\mathcal{M}_{\text{VAR}}$, and $\mathcal{M}_{\text{MAX}}$ as practical solutions). In Fig. 5 (left), we see that with only twelve trajectories, RE is able to reliably match expert performance for all oracles considered, while MM is not. The environment considered in this experiment is nearly deterministic, indicating that RE can help even when the environment is not stochastic. We hypothesize that the randomness in the initial state is sufficient for replay estimation to generate a significant improved estimate of the state-action visitation measure. This improvement is especially interesting considering both of the examples we study in Sec. 3 were heavily stochastic. We see a similar result in Fig. 5 (center), where we add in i.i.d. noise to the environment dynamics at each timestep. This makes the problem significantly more challenging than the standard version of the Walker task. RE is still able to match expert performance, with $\mathcal{M}_{\text{MAX}}$ working notably well. The correlation plot in Fig. 5 (right) shows us $\mathcal{M}_{\text{MAX}}$ appears to be best correlated with the idealized prefix weights, $\mathcal{M}_{\text{EXP}}$ under the state distribution induced by BC . Because of its superior performance, we use $\mathcal{M}_{\text{MAX}}$ for the rest of our experiments. In the left half of Fig. 6, we see RE improve the performance of MM . In the right half, we see RE out-perform BC in responding to an *extremely tiny* amount of noise added to the initial velocity of the agent (similar to the experiments of Reddy et al. [2019]) – we defer more details to Appendix C. These results indicate that RE can out-perform MM and BC , agreeing with our theory. We release our code at `https://github.com/gkswamy98/replay_est`. [2]

---

[2]After the initial publication of this paper, we continued to tune our baseline and fix bugs in our method's implementation. We then updated the result plots for both of the Noisy environments. Compared to their original performances, MM scores higher and RE scores lower. However, the RE still significantly out-performs MM.

# 6    Acknowledgements

ZSW is supported in part by the NSF FAI Award #1939606, a Google Faculty Research Award, a J.P. Morgan Faculty Award, a Facebook Research Award, an Okawa Foundation Research Grant, and a Mozilla Research Grant. GS is supported by a GPU award from NVIDIA. NR and JJ were partially supported by NSF Grants IIS-1901252, and CCF-1909499. KR is supported by ARO fund 051242-002.

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
