# A Proofs

## A.1 Notation

In this appendix, we use the notation $d_t^\pi(\cdot, \cdot)$ to indicate the state-action visitation measure induced by the policy $\pi$ at time $t$. We overload the notation $d_t^\pi(\cdot)$ to denote the state-visitation measure induced by the policy $\pi$ at time $t$. Likewise, the notations $d_t^D(\cdot, \cdot)$ and $d_t^D(\cdot)$ indicate the empirical visitation measures in the dataset $D$. For a function $g : \mathcal{X} \to \mathbb{R}$, the norm $\|g\|_\infty \triangleq \sup_{x \in \mathcal{X}} |g(x)|$.

Before discussing the proofs of the results, we also explain the instantiation of the function class in the tabular setting below.

**Remark 3.** *In the tabular setting, we instantiate the discriminator class as $\mathcal{F}_t = \{f_t : \|f_t\|_\infty \leq 1\}$ for each $t$, as the set of all $1$-bounded functions, and the policy class $\Pi$ as the set of all policies. eq. (4) corresponds to finding a policy which best matches the empirical state-action visitation measure observed in the dataset $D$ in total variation (TV) distance (see Appendix A.2 for a proof).*

## A.2 Imitation gap upper bound on empirical moment matching (Theorem 3.1)

Below we restate Theorem 3.1 and provide a proof of this result. The key observation is that since the learner $\pi^{\text{MM}}$ best matches the empirical distribution in the dataset, which is in turn close to the population visitation measure induced by $\pi^E$, we can expect the visitation measure induced by $\pi^E$ and $\pi^{\text{MM}}$ to be close. This in turns implies that both policies will collect a similar value under any reward function. Precisely characterizing the rates at which these distributions converge to one another results in the final bound.

**Theorem 3.1.** *Consider the empirical moment matching learner $\pi^{\text{MM}}$ (eq. (4)), instantiated with an appropriate discriminator class $\mathcal{F}$. The imitation gap satisfies $\mathbb{E}\left[J(\pi^E) - J(\pi^{\text{MM}})\right] \lesssim H\sqrt{|\mathcal{S}|/N_{\text{exp}}}$.*

*Proof.* Recall that the learner $\pi^{\text{MM}}$ is the solution to the following optimization problem,

$$\arg\min_\pi \sup_{f \in \mathcal{F}} \left\{ \mathbb{E}_\pi \left[ \frac{\sum_{t=1}^H f_t(s_t, a_t)}{H} \right] - \mathbb{E}_D \left[ \frac{\sum_{t=1}^H f_t(s_t, a_t)}{H} \right] \right\} \tag{19}$$

Exchanging the summation and maximization operators and recalling from Remark 3 that in the tabular setting, the discriminator class $\mathcal{F}$ is instantiated as the set of all $1$-bounded functions $\bigoplus_{t=1}^H \{f_t : \|f_t\|_\infty \leq 1\}$, $\pi^{\text{MM}}$ is a solution to

$$\arg\min_\pi \frac{1}{H} \sum_{t=1}^H \left( \sup_{f : \|f\|_\infty \leq 1} \mathbb{E}_\pi[f_t(s_t, a_t)] - \mathbb{E}_D[f_t(s_t, a_t)] \right) = \arg\min_\pi \frac{1}{H} \sum_{t=1}^H \text{TV}\left(d_t^\pi, d_t^D\right) \tag{20}$$

where the equation follows by the variational definition of the total variation distance, and where $d_t^\pi$ is the state-action visitation measure induced by $\pi^E$ and $d_t^D$ is the empirical state-action visitation measure in the dataset $D$. The imitation gap of this policy can be upper bounded by,

$$J(\pi^E) - J(\pi^{\text{MM}}) = \mathbb{E}_{\pi^E} \left[ \sum_{t=1}^H r_t(s_t, a_t) \right] - \mathbb{E}_{\pi^{\text{MM}}} \left[ \sum_{t=1}^H r_t(s_t, a_t) \right] \tag{21}$$

$$\stackrel{(i)}{\leq} \sum_{t=1}^H \sup_{r_t : \|r_t\|_\infty \leq 1} \left( \mathbb{E}_{\pi^E}[r_t(s_t, a_t)] - \mathbb{E}_{\pi^{\text{MM}}}[r_t(s_t, a_t)] \right) \tag{22}$$

$$\stackrel{(ii)}{=} \sum_{t=1}^H \text{TV}\left( d_t^{\pi^E}(\cdot, \cdot), d_t^{\pi^{\text{MM}}}(\cdot, \cdot) \right) \tag{23}$$

where $(i)$ maximizes over the reward function which is assumed to lie in the interval $[0, 1]$ pointwise. $(ii)$ again follows from the variational definition of total variation distance. This goes to show that in the tabular setting, MM is equivalent to finding the policy which best matches (in TV-distance) the empirical state-action distribution observed in the dataset.

By an application of triangle inequality,

$$J(\pi^E) - J(\pi^{\mathtt{MM}}) \leq \sum_{t=1}^{H} \mathsf{TV}\left(d_t^{\pi^E}(\cdot,\cdot), d_t^D(\cdot,\cdot)\right) + \mathsf{TV}\left(d_t^D(\cdot,\cdot), d_t^{\pi^{\mathtt{MM}}}(\cdot,\cdot)\right) \tag{24}$$

$$\leq 2\sum_{t=1}^{H} \mathsf{TV}\left(d_t^{\pi^E}(\cdot,\cdot), d_t^D(\cdot,\cdot)\right) \tag{25}$$

where $(i)$ follows from eq. (20) which shows that $\pi^{\mathtt{MM}}$ is the policy which best approximates the empirical state-action visitation measure in total variation distance, and therefore $\mathsf{TV}\left(d_t^{\pi^{\mathtt{MM}}}(\cdot,\cdot), d_t^D(\cdot,\cdot)\right) \leq \mathsf{TV}\left(d_t^{\pi^E}(\cdot,\cdot), d_t^D(\cdot,\cdot)\right)$. The final element is to identify the rate of convergence of the empirical visitation measure $d_t^D$, to the population $d_t^{\pi^E}$ in total variation distance. This result is known from Theorem 1 of Han et al. [2015], which shows that $\mathbb{E}\left[\mathsf{TV}\left(d_t^{\pi^E}(\cdot,\cdot), d_t^D(\cdot,\cdot)\right)\right] \lesssim \sqrt{\frac{|\mathcal{S}|}{N_{\exp}}}$ noting that $d_t^{\pi^E}$ is a distribution with support size $|\mathcal{S}|$ since $\pi^E$ is deterministic. Putting it together with eq. (25) after taking expectations on both sides gives,

$$J(\pi^E) - J(\pi^{\mathtt{MM}}) \lesssim \sum_{t=1}^{H} \sqrt{\frac{|\mathcal{S}|}{N_{\exp}}} = H\sqrt{\frac{|\mathcal{S}|}{N_{\exp}}}. \tag{26}$$

This completes the proof of the result. $\qquad\square$

### A.3   Lower bounding the Imitation gap of Behavior Cloning

Since BC is an offline algorithm - namely, the learner does not interact with the MDP, any lower bound against offline algorithms applies for behavior cloning as well. The lower bound instance in Rajaraman et al. [2020] is one such example. One state in the MDP is labelled as the "bad" state, $b$, which is absorbing and offers no reward. The remaining states each have a single "good" action which re-initializes the policy in a particular distribution $\rho$ on the set of good states, and offering a reward of $1$. The other actions at these states are "bad" and transition the learner to the bad state $b$ with probability $1$.

The key idea in the lower bound is that any offline algorithm does not know (i) which action the expert would have chosen at states unvisited in the dataset, and (ii) which action does what at these states. At best, the learner can correctly guess the good actions at a state with probability $1/|\mathcal{A}| \leq 1/2$. So, at each state unvisited in the dataset, the learner has a constant probability of getting stuck at the bad state in the MDP and collecting no reward then on. On the other hand, the expert would never choose bad actions at states and collects the maximum possible reward. By carefully counting the probability mass on the unvisited states, the expected imitation gap of any offline IL algorithm (such as BC ) can be lower bounded by $\Omega(|\mathcal{S}|H^2/N_{\exp})$ on these instances. Thus we have the following theorem,

**Theorem 4** (Theorem 6.1 of Rajaraman et al. [2020]). *Consider any learner $\widehat{\pi}$ which carries out an offline imitation learning algorithm (such as behavior cloning). Then, there exists an MDP instance such that the expected imitation gap is lower bounded by,*

$$\mathbb{E}\left[J(\pi^E) - J(\widehat{\pi})\right] \gtrsim \min\left\{H, \frac{|\mathcal{S}|H^2}{N_{\exp}}\right\}. \tag{27}$$

### A.4   Lower bounding the imitation gap of Empirical Moment Matching

In this section, we show that in the tabular setting, empirical moment matching is suboptimal compared for imitation learning in the worst-case. The main result we prove in this section is,

**Theorem 3.2.** *If $H \geq 4$, there is a tabular IL instance with $2$ states and actions on which with constant probability, the empirical moment matching learner (eq. (4)) incurs, $J(\pi^E) - J(\pi^{\mathtt{MM}}) \gtrsim H/\sqrt{N_{\exp}}$.*

**Remark 4.** *It is known from Rajaraman et al. [2020] that the Mimic-MD algorithm achieves an imitation gap of $\min\left\{H, \frac{\mathcal{S}|H^{3/2}}{N_{\exp}}, H\sqrt{\frac{|\mathcal{S}|}{N_{\exp}}}\right\}$. This is always better than the worst case error bound incurred by TV distribution matching from Theorem 3.2. In fact when $N_{\exp} \gtrsim \sqrt{H}$ the bound $\frac{H^{3/2}}{N_{\exp}}$ is significantly better than $\frac{H}{\sqrt{N_{\exp}}}$ which decays as $1/\sqrt{N_{\exp}}$. This is illustrated in Figure 4 .*

First note that the learner $\pi^{\mathrm{MM}}$ carries out empirical moment matching (eq. (4)), with the discriminator class $\mathcal{F}$ as initialized in Remark 3. As shown in eq. (20), the empirical moment matching learner can be redefined as the solution to a distribution matching problem,

$$\arg\min_{\pi} \frac{1}{H} \sum_{t=1}^{H} \mathsf{TV}\left(d_t^{\pi}(\cdot,\cdot), d_t^{D}(\cdot,\cdot)\right) \tag{28}$$

Consider an MDP instance with 2 states and 2 actions with a non-stationary transition and reward structure as described in Figure fig. 7. State 1 effectively has a single action (i.e. two actions, $a_1$ and $a_2$ with both inducing the same next-state distribution and reward). One of the actions at state 2 induces the uniform distribution over next states. The other action deterministically keeps the learner at state 2. The reward function is 0 at $t = 1$, and the action $a_2$ at state 2 is the only one which offers a reward of 1. The initial state distribution is highly skewed toward the state $s = 1$ and places approximately $1/\sqrt{N_{\mathrm{exp}}}$ mass on $s = 2$ and the remaining on $s = 1$.

Figure 7: MDP instance which shows that $L_1$ distribution matching is suboptimal. Here the transition structure is illustrated for $t = 1$. Both states have one action which reinitializes in the uniform distribution. State 2 has an additional action which keeps the state the same. The reward function is 0 for $t = 1$. For $t \geq 2$ the transition function is absorbing at both states; the reward function equals 1 at the state $s = 1$ for any action and is 0 everywhere else.

**MDP transition:** The state 2 is the only one with two actions. Action $a_1$ induces the uniform distribution over states, while action $a_2$ transitions the learner to state 2 with probability 1. Namely,

$$P_1(\cdot|s = 1, a) = \mathrm{Unif}(\mathcal{S}) \text{ for all } a \in \mathcal{A} \tag{29}$$
$$P_1(\cdot|s = 2, a_1) = \mathrm{Unif}(\mathcal{S}) \tag{30}$$
$$P_1(\cdot|s = 2, a_2) = \delta_2 \tag{31}$$

From time $t = 2$ onward, the actions are all absorbing. Namely, for all $t \geq 2$, $s \in \mathcal{S}$ and $a \in \mathcal{A}$,

$$P_t(\cdot|s, a) = \delta_s. \tag{32}$$

**Initial state distribution:** The initial state distribution $\rho = \left(1 - \frac{1}{\sqrt{N_{\mathrm{exp}}}}, \frac{1}{\sqrt{N_{\mathrm{exp}}}}\right)$.

**MDP reward function:** The reward function of the MDP encourages the learner to stay at the state $s = 1$ from time $t = 2$ onward. Namely,

$$r_t(s, a) = \begin{cases} 1, & \text{if } t \geq 2 \text{ and } s = 1 \\ 0, & \text{otherwise.} \end{cases} \tag{33}$$

**Expert policy:** At both states in the MDP, the expert picks the action $a_1$ to play, which induces the uniform distribution over actions at the next state. Namely, for each $t \in [H]$ and $s \in \mathcal{S}$,

$$\pi_t^E(\cdot|s) = \delta_{a_1} \tag{34}$$

The intuition behind the lower bound is as follows. The only action which affects the value of a policy is the choice made at $s = 2$ at time $t = 1$. At all other states, we may assume that there is effectively only a single action.

By the absorbing nature of states for $t \geq 2$, it turns out that if the observed empirical distribution in the dataset at time 2 is skewed toward state 2 (which is possible because of the inherent randomness in the data generation process), the learner's behavior at time 1 may be to *ignore* the expert's action observed at state $s = 2$, and instead pick the action $a_2$ which moves the learner to the state $s = 2$ deterministically. The learner is willing to choose a different action because the loss function eq. (28) encourages the state-action distribution at time $t = 2$ also to be well matched with what is observed

in the dataset. Even if it comes at the cost of picking an action different from what the expert plays. By exploiting this fact, we are able to show that the error incurred by a learner which solves eq. (28) in this simple 2 state example must be $\Omega(H/\sqrt{N_{\exp}})$.

Formally, we define 3 events,

(i) $\mathcal{E}_1$: All states in the MDP are visited in the dataset at each time $t = 1, 2, \cdots, H$.

(ii) $\mathcal{E}_2$: State 2 is visited at most $\sqrt{N_{\exp}}$ times at time 1 in the dataset $D$. In other words, $d_1^D(s = 2) = \delta'$ where $\delta' \leq \frac{1}{\sqrt{N_{\exp}}}$.

(iii) $\mathcal{E}_3$: At time 2 in the dataset $D$, the empirical distribution over states is of the form $\left(\frac{1}{2} - \delta, \frac{1}{2} + \delta\right)$ for some $\delta \geq \frac{2}{\sqrt{N_{\exp}}}$.

**Lemma 1.** *Jointly, the events $\mathcal{E}_1, \mathcal{E}_2$ and $\mathcal{E}_3$ occur with at least constant probability.*

$$\Pr(\mathcal{E}_1 \cap \mathcal{E}_2 \cap \mathcal{E}_3) \geq C, \tag{35}$$

*for some constant $C > 0$.*

*Proof.* By the absorbing nature of states for $t \geq 2$, it suffices for both states of the MDP to be visited in the dataset at time $t = 1, 2$. At time $t = 2$, the marginal state distribution under $\pi^E$ is the uniform distribution. By binomial concentration, both states are observed in the dataset at time $t = 2$ with probability $\geq 1 - e^{-C_1 N_{\exp}}$ for some constant $C_1 > 0$. On the other hand, at time $t = 1$, the marginal state distribution is $\rho = \left(1 - \frac{1}{\sqrt{N_{\exp}}}, \frac{1}{\sqrt{N_{\exp}}}\right)$. Yet again, by binomial concentration, both states are observed with probability $\geq 1 - e^{-C_2\sqrt{N_{\exp}}}$ for some constant $C_2 > 0$. By union bounding,

$$\Pr(\mathcal{E}_1) \geq 1 - e^{-C_1 N_{\exp}} - e^{-C_2 \sqrt{N_{\exp}}}. \tag{36}$$

Next we study $\mathcal{E}_2$ and $\mathcal{E}_3$ together. First of all, note that the state observed at $t = 1$ and $t = 2$ in a rollout of the expert policy are independent. This is because at both states at $t = 1$, the next state distribution under $\pi^E$ is uniform. Because of this fact, $\mathcal{E}_2$ and $\mathcal{E}_3$ are independent. Next we individually bound the probability of the two events.

$\mathcal{E}_2$: The number of times $s = 2$ is the initial state in trajectories the dataset $D$ is distributed as a binomial random variable with distribution $\text{Bin}(N_{\exp}, q)$ with $q = \rho(s = 2) = \frac{1}{\sqrt{N_{\exp}}}$. A median of a binomial random variable is $N_{\exp}q = \sqrt{N_{\exp}}$ (in fact any number in the interval $[\lfloor N_{\exp}q \rfloor, \lceil N_{\exp}q \rceil]$ is a median). Therefore, the probability that $s = 2$ is visited $\leq \sqrt{N_{\exp}}$ times in the dataset at time 1 is at least $1/2$. In summary,

$$\Pr(\mathcal{E}_2) \geq \frac{1}{2} \tag{37}$$

$\mathcal{E}_3$: The marginal distribution over states at time 2 in the dataset is uniform. Therefore, we expect the states 1 and 2 to be visited roughly $N_{\exp}/2$ times each in the dataset, but with a random variation of $\approx \sqrt{N_{\exp}}$ around this average. In other words, the empirical distribution fluctuates as $\left(\frac{1}{2} - \delta, \frac{1}{2} + \delta\right)$ with $\delta \geq \frac{2}{\sqrt{N_{\exp}}}$ with constant probability.

By the independence of $\mathcal{E}_2$ and $\mathcal{E}_3$ and union bounding to account for $\mathcal{E}_1$, the statement of the lemma follows. $\qquad\square$

**Lemma 2.** *For each $t \geq 2$,*

$$\mathsf{TV}(d_t^\pi(\cdot, \cdot), d_t^D(\cdot, \cdot)) \geq \mathsf{TV}(d_2^\pi(\cdot), d_2^D(\cdot)). \tag{38}$$

*The RHS is the TV distance between the state-visitation measure at time $t = 2$ under $\pi$ and that empirically observed in the dataset $D$. Conditioned on the events $\mathcal{E}_1, \mathcal{E}_2$ and $\mathcal{E}_3$ occuring, equality is met in eq. (38) if any only if $\pi_t(\cdot|s) = \pi_t^E(\cdot|s)$ for all states $s \in \mathcal{S}$.*

*Proof.* For any state $s \in \mathcal{S}$ and $t \geq 2$, observe that,

$$\sum_{a \in \mathcal{A}} \left| d_t^\pi(s, a) - d_t^D(s, a) \right| \tag{39}$$

$$= d_t^\pi(s)(1 - \pi_t(a^*|s)) + \left| d_t^\pi(s)\pi_t(a^*|s) - d_t^D(s, a^*) \right|, \text{ where } a^* = \pi_t^E(s), \tag{40}$$

$$\overset{(i)}{=} d_2^\pi(s)(1 - \pi_t(a^*|s)) + \left| d_2^\pi(s)\pi_t(a^*|s) - d_2^D(s, a^*) \right| \tag{41}$$

$$\overset{(ii)}{\geq} \left| d_2^\pi(s) - d_2^D(s) \right|, \tag{42}$$

where $(i)$ follows by the fact that the states of the MDP are absorbing under $\pi$ for $t \geq 2$. $(ii)$ follows by triangle inequality and using the fact that $\pi^E$ is deterministic, so $d_t^D(s, a^*) = d_t^D(s)$. Equality is met only if $\pi_t(a^*|s) = 1$ (since $d_t^D(s, a^*) > 0$ conditioned on $\mathcal{E}_1$). $\square$

The above lemma asserts the behavior of $\pi^{\text{MM}}$ in eq. (28) for $t \geq 2$. Namely, conditioned on the event $\mathcal{E}_1$ which happens with very high probability, all states are visited in the MDP and therefore, $\pi_t^{\text{MM}}(\cdot|s) = \pi_t^E(\cdot|s)$ for each state $s \in \mathcal{S}$ and time $t \geq 2$.

The only thing left to study is the MM learner's behavior at $t = 1$. We wish to show that with constant probability, the learner may choose to deviate from the expert policy in order to better match empirical state-action visitation measures. Conditioned on $\mathcal{E}_1$, the learner's policy at time $t = 1$ can be computed by solving the following optimization problem,

$$\text{TV}(d_1^\pi(\cdot, \cdot), d_1^D(\cdot, \cdot)) + (H - 1)\text{TV}(d_2^\pi(\cdot), d_2^D(\cdot)). \tag{43}$$

This follows directly by simplifying the learner's objective using Lemma 2.

Now, conditioned on the event $\mathcal{E}_1$, at time $t = 1$, the learner policy only needs to be optimized at the state $s = 2$. At the state $s = 1$, we may assume that the learner picks the expert's action $\pi_1^E(s = 1)$. To this end, suppose the learner picks the action $a_1$ with probability $p$ and the action $a_2$ with probability $1 - p$.

$$\text{TV}(d_1^\pi(\cdot, \cdot), d_1^D(\cdot, \cdot)) = \sum_{a \in \mathcal{A}} \left| d_1^{\pi^E}(s = 2, a) - d_1^D(s = 2, a) \right| \tag{44}$$

$$= |\rho(2)p - \delta'| + |\rho(2)(1 - p) - 0| \tag{45}$$

$$= \left| \frac{p}{\sqrt{N_{\text{exp}}}} - \delta' \right| + \frac{1 - p}{\sqrt{N_{\text{exp}}}}. \tag{46}$$

which follows by plugging in $\rho(2) = \frac{1}{\sqrt{N_{\text{exp}}}}$. And,

$$\text{TV}(d_2^\pi(\cdot), d_2^D(\cdot)) = \left| \left( \frac{1}{2} - \delta \right) - \frac{\rho(1)}{2} - \rho(2)\frac{p}{2} \right| + \left| \left( \frac{1}{2} + \delta \right) - \frac{\rho(1)}{2} - \rho(2)\left( (1 - p) + \frac{p}{2} \right) \right|. \tag{47}$$

Plugging in $\rho(2) = \frac{1}{\sqrt{N_{\text{exp}}}}$ and $\rho(1) = 1 - \frac{1}{\sqrt{N_{\text{exp}}}}$, we get,

$$\text{TV}(d_2^\pi(\cdot), d_2^D(\cdot)) = \left| \frac{1}{2\sqrt{N_{\text{exp}}}} - \delta - \frac{p}{2\sqrt{N_{\text{exp}}}} \right| + \left| \frac{p}{2\sqrt{N_{\text{exp}}}} - \frac{1}{2\sqrt{N_{\text{exp}}}} + \delta \right|. \tag{48}$$

Summing up eqs. (46) and (48), $p$ minimizes,

$$\underbrace{\left| \frac{p}{\sqrt{N_{\text{exp}}}} - \delta' \right| + \frac{1 - p}{\sqrt{N_{\text{exp}}}}}_{(i)} + (H - 1) \underbrace{\left( \left| \frac{p}{2\sqrt{N_{\text{exp}}}} + \delta - \frac{1}{2\sqrt{N_{\text{exp}}}} \right| + \left| \frac{1}{2\sqrt{N_{\text{exp}}}} - \delta - \frac{p}{2\sqrt{N_{\text{exp}}}} \right| \right)}_{(ii)}. \tag{49}$$

Intuitively, term $(i)$ captures the error incurred by the learner in the loss eq. (28) by deviating from $\pi^E$ at the first time step. Term $(ii)$ captures the decrease in the error at every subsequent time step because of the same deviation, since the learner is able to better match the state distribution at future time steps. In the next lemma we show that under events that hold with at least constant probability, the empirical moment matching learner chooses to play the wrong action at time $t = 1$ at the state $s = 2$.

**Lemma 3.** *Conditioned on the events $\mathcal{E}_2$ and $\mathcal{E}_3$, for $H \geq 4$, the unique minimizer of eq. (49) for $p \in [0, 1]$ is $p = 0$.*

*Proof.* The first term of eq. (49) is $\left| \frac{p}{\sqrt{N_{\exp}}} - \delta' \right| + \frac{1-p}{\sqrt{N_{\exp}}}$, the error from not picking the expert's action at state 1 at time 1 decreases at most linearly with a slope of $\frac{2}{\sqrt{N_{\exp}}}$.

Conditioned on the event $\mathcal{E}_3$, $\delta \geq \frac{2}{\sqrt{N_{\exp}}}$. Therefore, $\left| \frac{p}{2\sqrt{N_{\exp}}} + \delta - \frac{1}{2\sqrt{N_{\exp}}} \right| = \frac{p}{2\sqrt{N_{\exp}}} + \delta - \frac{1}{2\sqrt{N_{\exp}}}$. Therefore, the decrease in error at future steps by deviating from $\pi^E$ at the time $t = 1$, term $(ii)$ in eq. (49) is,

$$2(H-1) \left( \frac{p}{2\sqrt{N_{\exp}}} + \delta - \frac{1}{2\sqrt{N_{\exp}}} \right) \tag{50}$$

which is an increasing function of $p$ with slope $\frac{H-1}{\sqrt{N_{\exp}}}$. For $H \geq 4$ and the argument from the previous paragraph, this implies that term $(ii)$ increases more rapidly in $p$ than the rate at which term $(i)$ decreases. Therefore, the minimizer must be $p = 0$. $\qquad\square$

Thus, we conclude from Lemmas 2 and 3 that conditioned on the events $\mathcal{E}_1, \mathcal{E}_2$ and $\mathcal{E}_3$, the learner $\pi^{\texttt{MM}}$ perfectly mimics $\pi^E$ at each time $t \geq 2$, but deviates from the action played by $\pi^E$ at the state $s = 1$ at time $t = 1$.

Finally, we bound the difference in value between $\pi^E$ and $\pi^{\texttt{MM}}$ induced because of this deviation under the reward eq. (33).

**Lemma 4.** *Under the events $\mathcal{E}_1, \mathcal{E}_2$ and $\mathcal{E}_3$, under the reward eq. (33), the empirical moment matching learner $\pi^{\texttt{MM}}$ incurs imitation gap,*

$$J(\pi^E) - J(\pi^{\texttt{MM}}) = \frac{H}{2\sqrt{N_{\exp}}}. \tag{51}$$

*Proof.* Recall that under the events $\mathcal{E}_1, \mathcal{E}_2$ and $\mathcal{E}_3$, the learner $\pi^{\texttt{MM}}$ is identical to $\pi^E$ except at the state $s = 2$ where they perfectly deviate from each other. The state distribution induced by $\pi^E$ at each time $t \geq 2$ is the uniform distribution over states $\left( \frac{1}{2}, \frac{1}{2} \right)$. On the other hand, for $t \geq 2$, the state distribution induced by $\pi^{\texttt{MM}}$ at each time $t \geq 2$ is $\left( \rho(1)\frac{1}{2}, \rho(1)\frac{1}{2} + \rho(2) \right) = \left( \frac{1-1/\sqrt{N_{\exp}}}{2}, \frac{1+1/\sqrt{N_{\exp}}}{2} \right)$. Since the reward function is 1 on state 1, the difference in value between the expert and learner policy is,

$$J(\pi^E) - J(\pi^{\texttt{MM}}) = \frac{H}{2} - H \left( \frac{1 - 1/\sqrt{N_{\exp}}}{2} \right) = \frac{H}{2\sqrt{N_{\exp}}}. \tag{52}$$

This completes the proof. $\qquad\square$

Since $\mathcal{E}_1, \mathcal{E}_2$ and $\mathcal{E}_3$ jointly occur with constant probability by Lemma 1, this completes the proof of Theorem 3.2.

## B    Imitation gap of $\texttt{RE}$ : Proof of Theorem 2

In this section, we discuss a proof of a more general version of Theorem 2, where $N_{\text{replay}}$ can be finite. We prove the following result,

**Theorem 5.** *Consider the policy $\pi^{RE}$ returned by Algorithm 1. Assume that $\pi^E \in \Pi$ and the ground truth reward function $r_t \in \mathcal{F}_t$, which is assumed to be symmetric ($f_t \in \mathcal{F}_t \iff -f_t \in \mathcal{F}_t$) and bounded (For all $f_t \in \mathcal{F}_t$, $\|f_t\|_\infty \leq 1$). Choose $|D_1|, |D_2| = \Theta(N_{\exp})$ and suppose $N_{\text{replay}} \to \infty$. With probability $\geq 1 - 3\delta$,*

$$J(\pi^E) - J(\pi^{RE}) \lesssim \mathcal{L}_1 + \mathcal{L}_2 + \mathcal{L}_3 + \frac{\log\left(F_{\max}H/\delta\right)}{N_{\exp}} \tag{53}$$

where $F_{\max} \triangleq \max_{t \in [H]} |\mathcal{F}_t|$, and,

$$\mathcal{L}_1 \triangleq H^2 \, \mathbb{E}_{\pi^E} \left[ \frac{\sum_{t=1}^H \mathcal{M}(s_t, t) \mathsf{TV} \left( \pi_t^E(\cdot|s_t), \pi_t^{BC}(\cdot|s_t) \right)}{H} \right], \tag{54}$$

$$\mathcal{L}_2 \triangleq H^{3/2} \sqrt{\frac{\log \left( F_{\max} H/\delta \right)}{N_{\exp}} \frac{\sum_{t=1}^H \mathbb{E}_{\pi^E} \left[ 1 - \mathcal{M}(s_t, t) \right]}{H}}, \tag{55}$$

*And,*

$$\mathcal{L}_3 \triangleq H \sqrt{\frac{\log(F_{\max} H/\delta)}{N_{\text{replay}}}} + \frac{H \log(F_{\max} H/\delta)}{N_{\text{replay}}}. \tag{56}$$

Recall that the learner carrying out replay estimation returns the policy which minimizes the loss $\sup_{f \in \mathcal{F}} J_f(\pi) - \widehat{E}(f)$ over policies $\pi$, where $J_f(\pi) \triangleq \mathbb{E}_\pi \left[ \frac{1}{H} \sum_{t=1}^H f_t(s_t, a_t) \right]$ and abbreviating the notation $f = (f_1, \cdots, f_H)$. Note that,

$$J(\pi^E) - J(\pi^{\text{RE}}) \overset{(i)}{\leq} \sup_{f \in \mathcal{F}} J_f(\pi^E) - J_f(\pi^{\text{RE}}) \tag{57}$$

$$\leq \sup_{f \in \mathcal{F}} \left| J_f(\pi^E) - \widehat{E}(f) \right| + \sup_{f \in \mathcal{F}} \left| \widehat{E}(f) - J_f(\pi^{\text{RE}}) \right| \tag{58}$$

$$\overset{(ii)}{\leq} 2 \sup_{f \in \mathcal{F}} \left| J_f(\pi^E) - \widehat{E}(f) \right|. \tag{59}$$

where $(i)$ uses the realizability assumption that the ground truth reward lies in $\mathcal{F}$, and $(ii)$ uses the fact that $\pi^{\text{RE}}$ is a minimizer of eq. (5) and the fact that $\mathcal{F}$ is symmetric (this implies that $\sup_{f \in \mathcal{F}} J_f(\pi^E) - \widehat{E}(f) = \sup_{f \in \mathcal{F}} -J_f(\pi^E) + \widehat{E}(f) = \sup_{f \in \mathcal{F}} \left| J_f(\pi^E) - \widehat{E}(f) \right|$).

Note that $\widehat{E}(f)$ can be decomposed into a sum of two parts,

$$\widehat{E}^{(1)}(f) = \mathbb{E}_{D_{\text{replay}}} \left[ \frac{1}{H} \sum_{t=1}^H f_t(s_t, a_t) \left( 1 - \mathcal{P}(s_{1...t-1}) \right) \right], \text{ and,} \tag{60}$$

$$\widehat{E}^{(2)}(f) = \mathbb{E}_{D_2} \left[ \frac{1}{H} \sum_{t=1}^H f_t(s_t, a_t) \left( 1 - \mathcal{P}(s_{1...t-1}) \right) \right] \tag{61}$$

Likewise, we can decompose $J_f(\pi^E)$ into two terms,

$$J_f^{(1)}(\pi^E) \triangleq \mathbb{E}_{\pi^E} \left[ \sum_{t=1}^H f_t(s_t, a_t) \mathcal{P}(s_{1...t-1}) \right], \text{ and} \tag{62}$$

$$J_f^{(2)}(\pi^E) \triangleq \mathbb{E}_{\pi^E} \left[ \sum_{t=1}^H f_t(s_t, a_t) \left( 1 - \mathcal{P}(s_{1...t-1}) \right) \right] \tag{63}$$

Then, from eq. (59),

$$J(\pi^E) - J(\pi^{\text{RE}}) \leq 2 \sup_{f \in \mathcal{F}} \left| J_f(\pi^E) - \widehat{E}(f) \right| \tag{64}$$

$$\leq \underbrace{2 \sup_{f \in \mathcal{F}} \left| J_f^{(1)}(\pi^E) - \mathbb{E} \left[ \widehat{E}^{(1)}(f) \Big| D_1 \right] \right|}_{\text{(I)}} + \underbrace{2 \sup_{f \in \mathcal{F}} \left| \mathbb{E} \left[ \widehat{E}^{(1)}(f) \Big| D_1 \right] - \widehat{E}^{(1)}(f) \right|}_{\text{(II)}}$$

$$+ \underbrace{2 \sup_{f \in \mathcal{F}} \left| J_f^{(2)}(\pi^E) - \widehat{E}^{(2)}(f) \right|}_{\text{(III)}}. \tag{65}$$

where the last line follows by triangle inequality. We bound each of these terms in the next 3 lemmas, starting with (I).

**Lemma 5.**

$$\sup_{f\in\mathcal{F}}\left|J_f^{(1)}(\pi^E)-\mathbb{E}\left[\widehat{E}^{(1)}(f)\Big|D_1\right]\right|\le H\sum_{h=1}^{H}\mathbb{E}_{\pi^E}\left[\mathcal{M}(s_h,h)\,TV\left(\pi_h^E(\cdot|s_h),\pi_h^{BC}(\cdot|s_h)\right)\right] \quad (66)$$

*Proof.* The proof of this result closely follows the supervised learning reduction of BC (cf. Ross and Bagnell [2010]). Note that,

$$\mathbb{E}\left[\widehat{E}^{(1)}(f)\Big|D_1\right]-V_f^{(1)}(\pi^E)=\sum_{t=1}^{H}\mathbb{E}_{\pi^{BC}}\left[f_t(s_t,a_t)\mathcal{P}(s_{1\dots t-1})\right]-\mathbb{E}_{\pi^E}\left[f_t(s_t,a_t)\mathcal{P}(s_{1\dots t-1})\right]. \quad (67)$$

Define $\pi^{(h)}$ as the policy which plays $\pi^E$ until (and including) time $h$ and $\pi^{BC}$ after time $h$. Then, by cascading,

$$\mathbb{E}_{\pi^{BC}}\left[f_t(s_t,a_t)\mathcal{P}(s_{1\dots t-1})\right]-\mathbb{E}_{\pi^E}\left[f_t(s_t,a_t)\mathcal{P}(s_{1\dots t-1})\right]$$

$$=\sum_{h=0}^{t-1}\mathbb{E}_{\pi^{(h)}}\left[f_t(s_t,a_t)\mathcal{P}(s_{1\dots t-1})\right]-\mathbb{E}_{\pi^{(h+1)}}\left[f_t(s_t,a_t)\mathcal{P}(s_{1\dots t-1})\right] \quad (68)$$

Define, the uncertainty weighted state visitation measure $d^{\mathcal{M}}$ and the uncertainty weighted look-forward reward $\rho^{\mathcal{M}}$ as follows,

$$d_{h+1}^{\mathcal{M}}(s')\triangleq\mathbb{E}_{\pi^E}\left[\mathbb{1}(s_{h+1}=s')\prod_{t'=1}^{h}\mathcal{M}(s_{t'},t')\right] \quad (69)$$

$$\rho_{h+1}^{\mathcal{M}}(s',a')\triangleq\mathbb{E}_{\pi^{BC}}\left[f_t(s_t,a_t)\prod_{t'=h+2}^{t}\mathcal{M}(s_{t'},t')\Bigg|s_{h+1}=s',a_{h+1}=a'\right] \quad (70)$$

By decomposing expectations along trajectories, using the fact that $\mathcal{P}(s_{1\dots t-1})=\prod_{t'=1}^{H}\mathcal{M}(s_{t'},t')$ some simplification results in the following equation,

$$\left|\mathbb{E}_{\pi^{(h)}}\left[f_t(s_t,a_t)\mathcal{P}(s_{1\dots t-1})\right]-\mathbb{E}_{\pi^{(h+1)}}\left[f_t(s_t,a_t)\mathcal{P}(s_{1\dots t-1})\right]\right| \quad (71)$$

$$=\left|\sum_{s'\in\mathcal{S}}\sum_{a'\in\mathcal{A}}d_{h+1}^{\mathcal{M}}(s')\mathcal{M}(s',h+1)\left(\pi_{h+1}^E(a'|s')-\pi_{h+1}^{BC}(a'|s')\right)\rho_{h+1}^{\mathcal{M}}(s',a')\right| \quad (72)$$

$$\overset{(i)}{\le}\sum_{s'\in\mathcal{S}}d_{h+1}^{\mathcal{M}}(s')\mathcal{M}(s',h+1)TV\left(\pi_{h+1}^E(\cdot|s'),\pi_{h+1}^{BC}(\cdot|s')\right) \quad (73)$$

$$=\mathbb{E}_{\pi^E}\left[\mathcal{M}(s_{h+1},h+1)TV\left(\pi_{h+1}^E(\cdot|s_{h+1}),\pi_{h+1}^{BC}(\cdot|s_{h+1})\right)\right]. \quad (74)$$

where $(i)$ uses the fact that the membership oracle is a function $\in[0,1]$ and $f$ is bounded and lies in the interval $[0,1]$ (which implies that $\rho^{\mathcal{M}}$ also lies in $[0,1]$ pointwise). Plugging into eq. (68) and subsequently into eq. (67) completes the proof. $\qquad\square$

Next we bound the 3rd term, (III). This follows by an application of Bernstein's inequality.

**Lemma 6.** *With probability $\ge 1-\delta$,*

$$\sup_{f\in\mathcal{F}}\left|J_f^{(2)}(\pi^E)-\widehat{E}^{(2)}(f)\right|\le H\sqrt{\frac{\log(F_{\max}H/\delta)\sum_{t=1}^{H-1}\mathbb{E}_{\pi^E}\left[1-\mathcal{M}(s_t,t)\right]}{N_{\exp}}}+\frac{H\log(F_{\max}H/\delta)}{N_{\exp}} \quad (75)$$

*Proof.* First observe that,

$$J_f^{(2)}(\pi^E)-\widehat{V}_f^{(2)}=\sum_{t=1}^{H}\mathbb{E}_{tr\sim\mathrm{Unif}(D_2)}\left[f_t(s_t,a_t)\left(1-\mathcal{P}(s_{1\dots t-1})\right)\right]-\mathbb{E}_{\pi^E}\left[f_t(s_t,a_t)\left(1-\mathcal{P}(s_{1\dots t-1})\right)\right] \quad (76)$$

For each $t$, note that $f_t(s_t, a_t)(1 - \mathcal{P}(s_{1\ldots t-1}))$ is bounded in the range $[0, 1]$. Therefore, invoking Bernstein's inequality, with probability $\geq 1 - \delta$,

$$\left| \mathbb{E}_{\text{Unif}(D_2)} \left[ f_t(s_t, a_t)(1 - \mathcal{P}(s_{1\ldots t-1})) \right] - \mathbb{E}_{\pi^E} \left[ f_t(s_t, a_t)(1 - \mathcal{P}(s_{1\ldots t-1})) \right] \right| \tag{77}$$

$$\lesssim \sqrt{\frac{\text{Var}_{\pi^E}(f_t(s_t, a_t)(1 - \mathcal{P}(s_{1\ldots t-1}))) \log(1/\delta)}{N_{\text{exp}}}} + \frac{\log(1/\delta)}{N_{\text{exp}}} \tag{78}$$

$$\leq \sqrt{\frac{\mathbb{E}_{\pi^E}[(f_t(s_t, a_t)(1 - \mathcal{P}(s_{1\ldots t-1})))^2] \log(1/\delta)}{N_{\text{exp}}}} + \frac{\log(1/\delta)}{N_{\text{exp}}} \tag{79}$$

$$\overset{(i)}{\leq} \sqrt{\frac{\mathbb{E}_{\pi^E}[f_t(s_t, a_t)(1 - \mathcal{P}(s_{1\ldots t-1}))] \log(1/\delta)}{N_{\text{exp}}}} + \frac{\log(1/\delta)}{N_{\text{exp}}} \tag{80}$$

$$\overset{(ii)}{\leq} \sqrt{\frac{\mathbb{E}_{\pi^E}[1 - \mathcal{P}(s_{1\ldots t-1})] \log(1/\delta)}{N_{\text{exp}}}} + \frac{\log(1/\delta)}{N_{\text{exp}}} \tag{81}$$

where $(i)$ uses the fact that $f_t(s_t, a_t)(1 - \mathcal{P}(s_{1\ldots t-1}))$ is bounded in the range $[0, 1]$, and $(ii)$ uses the fact that $0 \leq f_t(s_t, a_t) \leq 1$. Assuming $0 \leq x_i \leq 1$ for all $i \in [n]$, we have the inequality,

$$1 - \prod_{i=1}^{n} x_i \leq \sum_{i=1}^{n} 1 - x_i \tag{82}$$

Applying this to eq. (81) for $1 - \mathcal{P}(s_{1\ldots t-1}) = 1 - \prod_{t'=1}^{t-1} \mathcal{M}(s_{t'}, t')$, we have,

$$\left| \mathbb{E}_{\text{Unif}(D_2)} \left[ f_t(s_t, a_t)(1 - \mathcal{P}(s_{1\ldots t-1})) \right] - \mathbb{E}_{\pi^E} \left[ f_t(s_t, a_t)(1 - \mathcal{P}(s_{1\ldots t-1})) \right] \right| \tag{83}$$

$$\leq \sqrt{\frac{\sum_{t'=1}^{t-1} \mathbb{E}_{\pi^E}[1 - \mathcal{M}(s_{t'}, t')] \log(1/\delta)}{N_{\text{exp}}}} + \frac{\log(1/\delta)}{N_{\text{exp}}} \tag{84}$$

Therefore, by union bounding, with probability $\geq 1 - \delta/H$, simultaneously for every $f_t \in \mathcal{F}_t$,

$$\left| \mathbb{E}_{\text{Unif}(D_2)} \left[ f_t(s_t, a_t)(1 - \mathcal{P}(s_{1\ldots t-1})) \right] - \mathbb{E}_{\pi^E} \left[ f_t(s_t, a_t)(1 - \mathcal{P}(s_{1\ldots t-1})) \right] \right| \tag{85}$$

$$\lesssim \sqrt{\frac{\log(|\mathcal{F}_t|H/\delta) \sum_{t'=1}^{t-1} \mathbb{E}_{\pi^E}[1 - \mathcal{M}(s_{t'}, t')]}{N_{\text{exp}}}} + \frac{\log(|\mathcal{F}_t|H/\delta)}{N_{\text{exp}}}. \tag{86}$$

This implies that the maximum over $f_t$ of the LHS is upper bounded by the RHS. Union bounding over $t = 1, \cdots, H$ and plugging into eq. (76), we have that with probability $\geq 1 - \delta$,

$$\sup_{f \in \mathcal{F}} \left| J_f^{(2)}(\pi^E) - \widehat{V}_f^{(2)} \right| \tag{87}$$

$$\leq \sum_{t=1}^{H} \left| \mathbb{E}_{\text{Unif}(D_2)} \left[ f_t(s_t, a_t)(1 - \mathcal{P}(s_{1\ldots t-1})) \right] - \mathbb{E}_{\pi^E} \left[ f_t(s_t, a_t)(1 - \mathcal{P}(s_{1\ldots t-1})) \right] \right| \tag{88}$$

$$\lesssim H \sqrt{\frac{\log(F_{\max}H/\delta) \sum_{t=1}^{H-1} \mathbb{E}_{\pi^E}[1 - \mathcal{M}(s_t, t)]}{N_{\text{exp}}}} + \frac{H \log(F_{\max}H/\delta)}{N_{\text{exp}}}. \tag{89}$$

$\square$

**Lemma 7.** *With probability $\geq 1 - \delta$,*

$$\sup_{f \in \mathcal{F}} \left| \mathbb{E} \left[ \widehat{V}_f^{(1)} \Big| D_1 \right] - \widehat{V}_f^{(1)} \right| \lesssim H \sqrt{\frac{\log(F_{\max}H/\delta)}{N_{\text{replay}}}} + \frac{H \log(F_{\max}H/\delta)}{N_{\text{replay}}}. \tag{90}$$

*Proof.* The proof follows essentially the same structure as Lemma 6 by decomposing $\widehat{V}_f^{(1)}$ into a sum of $H$ terms of the form $f_t(s_t, a_t)\mathcal{P}(s_{1\ldots t-1})$, applying Bernstein's inequality to bound the deviation of each term from its mean and finally union bounding over the rewards $f_t \in \mathcal{F}_t$ to get the uniform bound over all discriminators $f \in \mathcal{F}$. $\square$

Putting together Lemmas 5 to 7 with eq. (65) completes the proof of Theorem 2.

## B.1 Bound in the tabular setting (Theorem 6)

In this section, we provide an upper bound on the imitation gap of RE in the tabular setting when the expert is a deterministic policy. This recovers the bound on the imitation gap for RE proved in Rajaraman et al. [2020].

**Theorem 6.** *Consider an appropriately initialized version of RE, and let the size of the replay dataset $N_{\text{replay}} \to \infty$. For any tabular IL instance with $H \geq 10$, with probability $\geq 1 - 3\delta$,*

$$J(\pi^E) - J(\pi^{RE}) \lesssim \min\left\{ \frac{|\mathcal{S}|H^{3/2}}{N_{\text{exp}}}, H\sqrt{\frac{|\mathcal{S}|}{N_{\text{exp}}}} \right\} \log\left( \frac{|\mathcal{S}|H}{\delta} \right). \tag{91}$$

Below we describe the implementation of RE corresponding to Theorem 6 in more detail.

The membership oracle we use in this setting for RE is defined below,

$$\mathcal{M}(s,t) = \begin{cases} 1 & \text{if } s \text{ is visited in } D_1 \text{ at time } t \\ 0 & \text{otherwise.} \end{cases} \tag{92}$$

The function class $\mathcal{F}$ which we use is identical to that for empirical moment matching, which is described in Remark 3.

Note that in the tabular setting, BC simply mimics the deterministic expert's actions at states visited in the dataset $D_1$ and plays an arbitrary deterministic action on the remaining states. As a consequence of this definition, if $\mathcal{M}(s,t) = 1 \iff \pi_t^{BC}(\cdot|s) = \pi_t^E(\cdot|s)$ and $\mathcal{M}(s,t) = 0$ otherwise. We instantiate the family of discriminators as in Remark 3, as $\mathcal{F} = \bigoplus_{t=1}^{H}\{f_t : \|f_t\|_\infty \leq 1\}$ and the set of policies $\Pi$ optimized over is chosen as the set of all deterministic policies. While the guarantee of Theorem 2 depends on $F_{\max} = \max_{t \in [H]} |\mathcal{F}_t|$ which is unbounded (or $\exp(|\mathcal{S}||\mathcal{A}|)$ by using a discretization of the reward space), note that we can improve the guarantee to effectively have $F_{\max} \approx \exp(|\mathcal{S}|)$ noting the structure of the set of discriminators. Looking into the proof of Theorem 2 we bring out this dependence below. We note that there are many ways of bringing out this dependence, including a careful net argument directly on top of the guarantee of Theorem 2. We simply present one such argument below.

The critical step where the finiteness of the set of discriminators $\mathcal{F}$ is used, is in union bounding the gap between the population and the empirical estimate of $f_t(s_t, a_t)\left(1 - \mathcal{P}(s_{1...t-1})\right)$ in eq. (84).

$$\left| \mathbb{E}_{\text{Unif}(D_2)}\left[ f_t(s_t, a_t)\left(1 - \mathcal{P}(s_{1...t-1})\right) \right] - \mathbb{E}_{\pi^E}\left[ f_t(s_t, a_t)\left(1 - \mathcal{P}(s_{1...t-1})\right) \right] \right| \tag{93}$$

In the next step of the proof of Theorem 2, we union bound over all $f_t \in \mathcal{F}_t$. However, note that for $\mathcal{F}_t = \{f_t : \|f_t\|_\infty \leq 1\}$, we have that,

$$\sup_{f_t:\|f_t\|_\infty \leq 1} \left| \mathbb{E}_{\text{Unif}(D_2)}\left[ f_t(s_t, a_t)\left(1 - \mathcal{P}(s_{1...t-1})\right) \right] - \mathbb{E}_{\pi^E}\left[ f_t(s_t, a_t)\left(1 - \mathcal{P}(s_{1...t-1})\right) \right] \right| \tag{94}$$

$$\overset{(i)}{=} \sum_{s \in \mathcal{S}} \sum_{a \in \mathcal{A}} \left| \mathbb{E}_{\text{Unif}(D_2)}\left[ \mathbb{I}(s_t = s, a_t = a)\left(1 - \mathcal{P}(s_{1...t-1})\right) \right] - \mathbb{E}_{\pi^E}\left[ \mathbb{I}(s_t = s, a_t = a)\left(1 - \mathcal{P}(s_{1...t-1})\right) \right] \right| \tag{95}$$

$$\overset{(ii)}{=} \sum_{s \in \mathcal{S}} \left| \mathbb{E}_{\text{Unif}(D_2)}\left[ \mathbb{I}(s_t = s)\left(1 - \mathcal{P}(s_{1...t-1})\right) \right] - \mathbb{E}_{\pi^E}\left[ \mathbb{I}(s_t = s)\left(1 - \mathcal{P}(s_{1...t-1})\right) \right] \right| \tag{96}$$

$$\overset{(iii)}{\leq} \sum_{s \in \mathcal{S}} \left| \mathbb{E}_{\text{Unif}(D_2)}\left[ \mathbb{I}(s_t = s)\left(1 - \mathcal{P}(s_{1...t-1})\right) \right] - \mathbb{E}_{\pi^E}\left[ \mathbb{I}(s_t = s)\left(1 - \mathcal{P}(s_{1...t-1})\right) \right] \right| \tag{97}$$

where $(i)$ follows similar to the equivalence between the variational representation of TV distance $(\text{TV}(P,Q) = \frac{1}{2}\sup_{f:\|f\|_\infty \leq 1} \mathbb{E}_P[f] - \mathbb{E}_Q[f])$ and the relationship to the $L_1$ distance, $\text{TV}(P,Q) = \frac{1}{2}L_1(P,Q)$. On the other hand, $(ii)$ follows by noting that the expert is a deterministic policy (and $D_2$ is generated by rolling out $\pi^E$). $(iii)$ follows by an application of Holder's inequality. By subgaussian

concentration, for each $s \in \mathcal{S}$, with probability $\geq 1 - \frac{\delta}{|\mathcal{S}|H}$,

$$\left| \mathbb{E}_{\text{Unif}(D_2)} \left[ \mathbb{I}(s_t = s)\left(1 - \mathcal{P}(s_{1\ldots t-1})\right) \right] - \mathbb{E}_{\pi^E} \left[ \mathbb{I}(s_t = s)\left(1 - \mathcal{P}(s_{1\ldots t-1})\right) \right] \right| \tag{98}$$

$$\lesssim \sqrt{\frac{\text{Var}_{\pi^E}\left(\mathbb{I}(s_t = s)\left(1 - \mathcal{P}(s_{1\ldots t-1})\right)\right) \log\left(\frac{|\mathcal{S}|H}{\delta}\right)}{|D_2|}} + \frac{\log\left(\frac{|\mathcal{S}|H}{\delta}\right)}{|D_2|} \tag{99}$$

$$\overset{(i)}{\leq} \sqrt{\frac{\mathbb{E}_{\pi^E}\left[\mathbb{I}(s_t = s)\left(1 - \mathcal{P}(s_{1\ldots t-1})\right)\right] \log\left(\frac{|\mathcal{S}|H}{\delta}\right)}{|D_2|}} + \frac{\log\left(\frac{|\mathcal{S}|H}{\delta}\right)}{|D_2|} \tag{100}$$

where $(i)$ uses the fact that $0 \leq \mathbb{I}(s_t = s)\left(1 - \mathcal{P}(s_{1\ldots t-1})\right) \leq 1$. Combining with eq. (97), union bounding and applying Cauchy Schwarz inequality, with probability $\geq 1 - \frac{\delta}{H}$,

$$\sup_{f_t: \|f_t\|_\infty \leq 1} \left| \mathbb{E}_{\text{Unif}(D_2)}\left[f_t(s_t, a_t)\left(1 - \mathcal{P}(s_{1\ldots t-1})\right)\right] - \mathbb{E}_{\pi^E}\left[f_t(s_t, a_t)\left(1 - \mathcal{P}(s_{1\ldots t-1})\right)\right] \right| \tag{101}$$

$$\lesssim \sqrt{|\mathcal{S}|} \sqrt{\sum_{s \in \mathcal{S}} \frac{\mathbb{E}_{\pi^E}\left[\mathbb{I}(s_t = s)\left(1 - \mathcal{P}(s_{1\ldots t-1})\right)\right] \log\left(\frac{|\mathcal{S}|H}{\delta}\right)}{|D_2|}} + \frac{|\mathcal{S}| \log\left(\frac{|\mathcal{S}|H}{\delta}\right)}{|D_2|} \tag{102}$$

$$= \sqrt{|\mathcal{S}|} \sqrt{\frac{\mathbb{E}_{\pi^E}\left[1 - \mathcal{P}(s_{1\ldots t-1})\right] \log\left(\frac{|\mathcal{S}|H}{\delta}\right)}{|D_2|}} + \frac{|\mathcal{S}| \log\left(\frac{|\mathcal{S}|H}{\delta}\right)}{|D_2|} \tag{103}$$

$$\overset{(i)}{\leq} \min\left\{ \sqrt{\frac{|\mathcal{S}| \log\left(\frac{|\mathcal{S}|H}{\delta}\right)}{|D_2|}}, \sqrt{|\mathcal{S}| \frac{\sum_{t=1}^{H-1} \mathbb{E}_{\pi^E}\left[1 - \mathcal{M}(s_t, t)\right] \log\left(\frac{|\mathcal{S}|H}{\delta}\right)}{|D_2|}} \right\} + \frac{|\mathcal{S}| \log\left(\frac{|\mathcal{S}|H}{\delta}\right)}{|D_2|} \tag{104}$$

where $(i)$ follows by the same simplification as in eq. (82). Comparing with eq. (86), this roughly corresponds to setting $F_{\max} \approx \exp(|\mathcal{S}|)$. All in all, summing eq. (104) over $t \in [H]$ and plugging into eq. (88), with probability $\geq 1 - \delta$,

$$\sup_{f \in \mathcal{F}} \left| J_f^{(2)}(\pi^E) - \widehat{V}_f^{(2)} \right| \tag{105}$$

$$\lesssim H\left\{ \sqrt{\frac{|\mathcal{S}| \log\left(\frac{|\mathcal{S}|H}{\delta}\right)}{|D_2|}}, \sqrt{|\mathcal{S}| \frac{\sum_{t=1}^{H-1} \mathbb{E}_{\pi^E}\left[1 - \mathcal{M}(s_t, t)\right] \log\left(\frac{|\mathcal{S}|H}{\delta}\right)}{|D_2|}} \right\} + \frac{|\mathcal{S}| \log\left(\frac{|\mathcal{S}|H}{\delta}\right)}{|D_2|} \tag{106}$$

Finally, we plug this into eq. (65), which is restated below,

$$J(\pi^E) - J(\pi^{\text{RE}}) \leq 2 \sup_{f \in \mathcal{F}} \left| J_f(\pi^E) - \widehat{E}(f) \right| \tag{107}$$

$$\leq 2 \underbrace{\sup_{f \in \mathcal{F}} \left| J_f^{(1)}(\pi^E) - \mathbb{E}\left[ \widehat{E}^{(1)}(f) \Big| D_1 \right] \right|}_{(I)} + 2 \underbrace{\sup_{f \in \mathcal{F}} \left| \mathbb{E}\left[ \widehat{E}^{(1)}(f) \Big| D_1 \right] - \widehat{E}^{(1)}(f) \right|}_{(II)} \tag{108}$$

$$+ 2 \underbrace{\sup_{f \in \mathcal{F}} \left| J_f^{(2)}(\pi^E) - \widehat{E}^{(2)}(f) \right|}_{(III)}. \tag{109}$$

For the chosen membership oracle in eq. (92), the term (I) is 0, since by Lemma 5 it is upper bounded by $H \sum_{h=1}^{H} \mathbb{E}_{\pi^E}\left[\mathcal{M}(s_h, h)\text{TV}\left(\pi_h^E(\cdot|s_h), \pi_h^{\text{BC}}(\cdot|s_h)\right)\right]$. This is equal to 0 since $\mathcal{M}(s, t) = 0$ wherever $\pi_t^E(\cdot|s) \neq \pi_t^{\text{BC}}(\cdot|s)$. On the other hand, $N_{\text{replay}} \to \infty$ ensures that the term (III) goes to

0 by the strong law of large numbers. Therefore, with probability $\geq 1 - 2\delta$,

$$J(\pi^E) - J(\pi^{\mathtt{RE}}) \tag{110}$$

$$\leq 2 \sup_{f \in \mathcal{F}} \left| \mathbb{E}\left[ \widehat{E}^{(1)}(f) \middle| D_1 \right] - \widehat{E}^{(1)}(f) \right| \tag{111}$$

$$\lesssim H \left\{ \sqrt{\frac{|\mathcal{S}| \log\left(\frac{|\mathcal{S}|H}{\delta}\right)}{|D_2|}}, \sqrt{|\mathcal{S}| \frac{\sum_{t=1}^{H-1} \mathbb{E}_{\pi^E}\left[1 - \mathcal{M}(s_t, t)\right] \log\left(\frac{|\mathcal{S}|H}{\delta}\right)}{|D_2|}} \right\} + \frac{|\mathcal{S}|H \log\left(\frac{|\mathcal{S}|H}{\delta}\right)}{|D_2|} \tag{112}$$

Finally, we bound $\mathbb{E}_{\pi^E}[1 - \mathcal{M}(s_t, t)]$ for the membership oracle defined in eq. (92). By definition, this quantity is the same as $\mathrm{Pr}_{\pi^E}(s_t$ not visited in $D_1$ at time t ). This is the probability that given $N_{\exp}$ samples from a distribution (the state visited at time $t$ in an expert rollout), the probability that a new sample from the same distribution is not in the support of the observed samples. This is known as the missing mass McAllester and Schapire [2000]. In Lemma A.3 Rajaraman et al. [2020] it is shown that with probability $\geq 1 - \delta$,

$$\sum_{t=1}^{H-1} \mathrm{Pr}_{\pi^E}(s_t \text{ not visited in } D_1 \text{ at time t}) \lesssim \frac{|\mathcal{S}|H}{|D_1|} + \frac{\sqrt{|\mathcal{S}|} H \log\left(\frac{|\mathcal{S}|H}{\delta}\right)}{|D_1|} \tag{113}$$

Finally, combining with eq. (112) and using the fact that that $|D_1|, |D_2| = \Theta(N_{\exp})$, with probability $\geq 1 - 3\delta$,

$$J(\pi^E) - J(\pi^{\mathtt{RE}}) \lesssim \min \left\{ H\sqrt{\frac{|\mathcal{S}| \log\left(\frac{|\mathcal{S}|H}{\delta}\right)}{N_{\exp}}}, \frac{|\mathcal{S}|H^{3/2}}{N_{\exp}} \log\left(\frac{|\mathcal{S}|H}{\delta}\right) \right\}. \tag{114}$$

This completes the proof of Theorem 6.

## B.2   Bound with parametric function approximation under Lipschitzness

In this section, we provide an upper bound on the imitation gap of $\mathtt{RE}$ in the presence of parametric function approximation under a Lipschitzness assumption on the function classes, and assuming access to a parameter estimation oracle for offline classification.

**Definition** (IL with function-approximation). *In this setting, for each $t \in [H]$, there is a parameter class $\Theta_t \subseteq \mathbb{B}_2^d$, the unit $L_2$ ball in $d$ dimensions, and an associated function class $\{f_{\theta_t} : \theta_t \in \Theta_t\}$. For each $t \in [H]$ there exists an unknown $\theta_t^E \in \Theta_t$ such that $\forall s \in \mathcal{S}$,*

$$\pi_t^E(s) = \arg\max_{a \in \mathcal{A}} f_{\theta_t^E}(s, a). \tag{115}$$

**Definition 3** (Policy induced by a classifier). *Consider a set of parameters $\theta = \{\theta_1, \cdots, \theta_H\}$ where $\theta_t \in \Theta_t$ for each $t$. A policy $\pi^\theta$ is said to be induced by the set of classifiers defined by $\theta$ if for all $s \in \mathcal{S}$ and $t \in [H]$,*

$$\pi_t^\theta(s) = \arg\max_{a \in \mathcal{A}} f_{\theta_t}(s, a). \tag{116}$$

*By this definition, $\pi^E = \pi^{\theta^E}$ where $\theta^E = \{\theta_1^E, \cdots, \theta_H^E\}$.*

**Definition 4** (Lipschitz parameterization). *A function class $\mathcal{G} = \{g_\theta : \theta \in \Theta\}$ where $g_\theta(\cdot) : \mathcal{X} \to \mathbb{R}$ is said to satisfy L-Lipschitz parameterization if, $\|g_\theta(\cdot) - g_{\theta'}(\cdot)\|_\infty \leq L\|\theta - \theta'\|_2$ for all $\theta, \theta' \in \Theta$. In other words, for each $x \in \mathcal{X}$, $g_\theta(x)$ is an L-Lipschitz function in $\theta$, in the $L_2$ norm.*

**Assumption 4** (Assumption 1 restated). *For each $t$, the class $\{f_{\theta_t} : \theta_t \in \Theta_t\}$ is L-Lipschitz in its parameterization, $\theta_t \in \Theta_t$.*

To deal with parametric function approximation, we assume that the learner has access to a proper offline classification oracle, which given a dataset of classification examples, guarantees to approximately return the underlying ground truth parameter. Namely,

**Assumption 5** (Assumption 2 restated). *We assume that the learner has access to a multi-class classification oracle, which given $n$ examples of the form, $(s^i, a^i)$ where $s^i \overset{i.i.d.}{\sim} \mathcal{D}$ and $a^i = \arg\max_{a \in \mathcal{A}} f_{\theta^*}(s^i, a)$, returns a $\hat{\theta} \in \Theta$ such that, with probability $\geq 1 - \delta$, $\|\hat{\theta} - \theta^*\|_2 \leq \mathcal{E}_{\Theta, n, \delta}$.*

This assumption implies that the parameter class $\Theta_t$ (and the associated function class $\{f_{\theta_t} : \theta_t \in \Theta_t\}$) admits finite sample complexity guarantees for learning the parameter $\theta_t^*$ given classification examples from the underlying ground truth function $f_{\theta_t^*}$. As we discuss in more detail later, we will assume that this classification oracle is used by RE to train the BC policy in Line 3 of Algorithm 1.

Finally, we introduce the main assumption on the IL instances we study. We assume that the classification problems solved by BC at each $t \in [H]$ satisfy a margin condition.

**Assumption 6** (Assumption 3 restated). *For $\theta \in \Theta_t$, define $a_s^\theta = \arg\max_{a \in \mathcal{A}} f_\theta(s, a)$ as the classifier output on the state $s$. The weak margin condition assumes that for each $t$, there is no classifier $\theta \in \Theta_t$ such that for a large mass of states, $f_\theta(s_t, a_{s_t}^\theta) - \max_{a \neq a_{s_t}^\theta} f_\theta(s_t, a)$, i.e. the "margin" from the nearest classification boundary, is small. Formally, the weak-margin condition with parameter $\mu$ states that, for any $\theta \in \Theta_t$ and $\eta \leq 1/\mu$,*

$$\Pr_{\pi^E} \left( f_\theta(s_t, a_{s_t}^\theta) - \max_{a \neq a_{s_t}^\theta} f_\theta(s_t, a) \geq \eta \right) \geq e^{-\mu\eta}. \tag{117}$$

*The weak margin condition only assumes that there is at least an exponentially small (in $\eta$) mass of states with margin at least $\eta$. Smaller $\mu$ indicates a larger mass away from any decision boundary. It suffices to assume that eq. (117) is only true for $\theta$ as the classifier in Assumption 5 for our guarantees (Theorem 7) to hold.*

**Remark 5.** *Note that the weak margin condition is the multi-class extension of the Tsybakov margin condition of Mammen and Tsybakov [1999], Audibert and Tsybakov [2007] defined for the binary case. In particular, in eq. (117), we may replace the RHS by $1 - \mu\eta$, or $1 - (\mu\eta)^\alpha$ for $\alpha > 0$ to get different analogs of the margin condition and the main guarantee, Theorem 3, as we discuss in Appendix B.2.*

**Theorem 7.** *For IL with parametric function approximation, under Assumptions 4 to 6, appropriately instantiating RE ensures that with probability $\geq 1 - 4\delta$,*

$$J(\pi^E) - J(\pi^{RE}) \lesssim H^{3/2} \sqrt{\frac{\mu L \log(F_{\max} H/\delta)}{N_{\exp}} \frac{\sum_{t=1}^H \mathcal{E}_{\Theta_t, N_{\exp}, \delta/H}}{H}} + \frac{\log(F_{\max} H/\delta)}{N_{\exp}}. \tag{118}$$

*Note that we assume the same conditions on $\mathcal{F}$ as required in Theorem 2.*

**Remark 6.** *The classification oracle in Assumption 5 asks for a stronger condition than just finding a classifier with small generalization error, which need not be close to the ground truth $\theta^*$ in the parameter space. Learning classifiers with small generalization error is studied in Daniely et al. [2013] who show that the Natarajan dimension, up to log-factors in the number of classes (i.e. number of actions) is the right statistical complexity measure which characterizes the generalization error of the best learner. In the realizable case, the optimal generalization error guarantee scales as $\widetilde{O}(1/n)$ where $n$ is the number of classification examples. Under certain assumptions on the input distribution $\mathcal{D}$ and the function family (e.g. for linear families), we later show that the generalization error guarantee can be extended to approximately learning the parameter as well (up to problem dependent constants). Generally, under two conditions,*

    *1. Generalization error guarantees which scale as $\widetilde{O}(1/N_{\exp})$ can be extended to parameter learning,*

    *2. The dependence of the generalization error on the failure probability $\delta$ scales as $\mathrm{polylog}(1/\delta)$,*

*the optimality gap for RE which we prove in Theorem 7 scales as $\widetilde{O}\left(H^{3/2}/N_{\exp}\right)$. The constants in $\widetilde{O}(\cdot)$ here depend on the Natarajan dimension and the covering number of the function classes $\mathcal{F}_t$ among problem dependent constants.*

In proving this result, we first discuss the implementation of RE .

**Implementation of** `RE` **(Algorithm 1)**   We discuss the instantiation of `RE` in the Lipschitz setting below. The underlying function class $\mathcal{F}$ is chosen arbitrarily (note that the guarantee we prove depends on this function class, and the only constraints on $\mathcal{F}$ are those in Theorem 2 - the ground truth reward must belong in $\mathcal{F} = \otimes_{t=1}^{H} \mathcal{F}_t$, the function class is symmetric, i.e., $f_t \in \mathcal{F}_t \iff -f_t \in \mathcal{F}_t$ for each $t$ and for all $f_t \in \mathcal{F}_t$, $\|f_t\|_{\infty} \le 1$) This requires specifying the choice of the membership oracle $\mathcal{M}$ and describing the instantiation of `BC` .

**Implementation of** `BC` **:**   Recall that in Algorithm 1, the learner trains `BC` on the dataset $D_1$. In particular, under the offline classification oracle condition, Assumption 5, the learner trains $H$ classifiers, one for each $t$, trained on the state-action pairs (i.e. state is the input, and the action at this state is the corresponding class) observed in the expert dataset at time $t$ using the offline classifier in Assumption 5. We assume that each classifier is trained with the failure probability chosen as $\delta/H$. Denoting this set of $H$ classifiers as $\widehat{\theta}^{\mathsf{BC}} = \left\{ \widehat{\theta}_1^{\mathsf{BC}}, \cdots, \widehat{\theta}_H^{\mathsf{BC}} \right\}$, this corresponds to the to the policy $\pi^{\mathsf{BC}} = \pi^{\widehat{\theta}^{\mathsf{BC}}}$ induced by the classifier $\widehat{\theta}^{\mathsf{BC}}$ (Definition 3).

In particular, by union bounding, the classifiers $\widehat{\theta}^{\mathsf{BC}}$ satisfy with probability $\ge 1 - \delta$ simultaneously for each time $t \in [H]$,

$$\|\theta_t^E - \hat{\theta}_t\|_2 \le \mathcal{E}_{\Theta_t, N_{\exp}, \delta/H}. \tag{119}$$

**Membership oracle:**   Fix a time-step $t \in [H]$. The membership oracle $\mathcal{M}$ is defined in eq. (10) as,

$$\mathcal{M}(s, t) = \begin{cases} +1 & \text{if } \exists a \in \mathcal{A} \text{ such that, } \forall a' \in \mathcal{A}, \ f_{\widehat{\theta}_t^{\mathsf{BC}}}(s, a) - f_{\widehat{\theta}_t^{\mathsf{BC}}}(s, a') \ge 2L\mathcal{E}_{\Theta_t, N_{\exp}, \delta/H} \\ 0 & \text{otherwise.} \end{cases} \tag{120}$$

We first show that on the states such that the membership oracle is 1, the expert policy perfectly matches the learner's policy.

**Lemma 8.** *At every state $s$ such that $\mathcal{M}(s, t) = +1$, $\pi_t^E(s) = \pi_t^{BC}(s)$.*

*Proof.* Note that $\theta_t^E$ satisfies $\|\theta_t^E - \widehat{\theta}_t^{\mathsf{BC}}\|_2 \le \mathcal{E}_{\Theta_t, N_{\exp}, \delta/H}$ with probability $1 - \delta$. Consider the action $a$ played by the learner, for any $a' \in \mathcal{A}$,

$$f_{\theta_t^E}(s, a) - f_{\theta_t^E}(s, a') \ge f_{\widehat{\theta}_t^{\mathsf{BC}}}(s, a) - \mathcal{E}_{\Theta_t, N_{\exp}, \delta/H} L - f_{\widehat{\theta}_t^{\mathsf{BC}}}(s, a') - \mathcal{E}_{\Theta_t, N_{\exp}, \delta/H} L \tag{121}$$

$$\ge 0 \tag{122}$$

where the first inequality follows by Lipschitzness of $f_{\cdot}(s, a)$ and the last inequality follows by definition of the set of states where $\mathcal{M}(s, t) = +1$: $\forall a' \in \mathcal{A}$, $f_{\widehat{\theta}_t^{\mathsf{BC}}}(s, a) - f_{\widehat{\theta}_t^{\mathsf{BC}}}(s, a') \ge 2\mathcal{E}_{\Theta_t, N_{\exp}, \delta/H} L$.

Since for this action $a$, $f_{\theta_t^E}(s, a) - f_{\theta_t^E}(s, a') \ge 0$ for all other actions $a' \in \mathcal{A}$, $a$ must be the action played by the expert policy. This completes the proof.   □

Note that $\pi^{\mathsf{BC}}$ always matches $\pi^E$ wherever the membership oracle $\mathcal{M}$ is non-zero. We run Algorithm 1. Therefore, from Theorem 2, with probability $\ge 1 - 4\delta$, the imitation gap of the learner is bounded by,

$$J_r(\pi^E) - J_r(\pi^{\mathsf{RE}}) \lesssim H^{3/2} \sqrt{\frac{\log\left(F_{\max} H/\delta\right)}{N_{\exp}} \frac{\sum_{t=1}^{H} \mathbb{E}_{\pi^E}\left[1 - \mathcal{M}(s_t, t)\right]}{H}} + \frac{\log\left(F_{\max} H/\delta\right)}{N_{\exp}}. \tag{123}$$

To complete the proof, we must bound $\mathbb{E}_{\pi^E}\left[1 - \mathcal{M}(s_t, t)\right]$, which is the measure of states $s$ such that $\forall a \in \mathcal{A}, \exists a' \in \mathcal{A} : f_{\hat{\theta}_t}(s, a) - f_{\hat{\theta}_t}(s, a') \le 2L\mathcal{E}_{\Theta_t, N_{\exp}, \delta/H}$, i.e. the mass of states which are very close to a decision boundary. The probability of this set of states is upper bounded by the weak margin condition. Indeed, for each $t \in [H]$, defining $a_s^* = \arg\max_{a \in \mathcal{A}} f_{\widehat{\theta}_t^{\mathsf{BC}}}(s, a)$,

$$\Pr_{\pi^E}\left(f_{\widehat{\theta}_t^{\mathsf{BC}}}(s_t, a_{s_t}^*) - \max_{a \ne a_{s_t}^*} f_{\widehat{\theta}_t^{\mathsf{BC}}}(s_t, a) \ge 2L\mathcal{E}_{\Theta_t, N_{\exp}, \delta/H}\right) \ge e^{-\mu L\mathcal{E}_{\Theta_t, N_{\exp}, \delta/H}} \tag{124}$$

$$\ge 1 - \mu L\mathcal{E}_{\Theta_t, N_{\exp}, \delta/H}. \tag{125}$$

Therefore,

$$\mathbb{E}_{\pi^E}\left[1 - \mathcal{M}(s_t, t)\right] \lesssim \mu L \mathcal{E}_{\Theta_t, N_{\exp}, \delta/H}. \tag{126}$$

Putting it together with eq. (123), and simplifying, with probability $\geq 1 - 4\delta$,

$$J(\pi^E) - J(\pi^{\mathtt{RE}}) \lesssim H^{3/2} \sqrt{\frac{\mu L \log\left(F_{\max} H/\delta\right) \sum_{t=1}^{H} \mathcal{E}_{\Theta_t, N_{\exp}, \delta/H}}{N_{\exp}} \frac{}{H}} + \frac{\log\left(F_{\max} H/\delta\right)}{N_{\exp}}. \tag{127}$$

Note that in Equation (125), we only use the fact that the probability mass of states which are $\eta$-close to any decision boundary is not too high. Similar to Audibert and Tsybakov [2007], we may consider relaxations of the weak margin condition, as below.

**Assumption 7** ($\alpha$-weak margin condition). *Consider any $t \in [H]$ and $\theta \in \Theta_t$. For each $s \in \mathcal{S}$, define $a_s^* = \arg\max_{a \in \mathcal{A}} f_\theta(s, a)$ as the classifier output under $f_\theta$. The $\alpha$ weak margin condition with parameter $\mu$ assumes that, for any $\eta \leq 1/\mu$,*

$$\forall \theta \in \Theta_t, \ \Pr_{\pi^E}\left(f_\theta(s_t, a_{s_t}^*) - \max_{a \neq a_{s_t}^*} f_\theta(s_t, a) \geq \eta\right) \geq 1 - (\mu\eta)^\alpha. \tag{128}$$

*When $\alpha = 1$, this condition is effectively equivalent to the weak margin condition in Assumption 6.*

Following the proof of Theorem 7, we may obtain the following result under the $\alpha$ weak margin condition for $\alpha \neq 1$.

**Theorem 8.** *For IL with parametric function approximation, under Assumptions 4, 5 and 7, appropriately instatiating* `RE` *ensures that with probability $\geq 1 - 4\delta$,*

$$J(\pi^E) - J(\pi^{\mathtt{RE}}) \lesssim H^{3/2} \sqrt{\frac{(\mu L)^\alpha \log\left(F_{\max} H/\delta\right) \sum_{t=1}^{H} (\mathcal{E}_{\Theta_t, N_{\exp}, \delta/H})^\alpha}{N_{\exp}} \frac{}{H}} + \frac{\log\left(F_{\max} H/\delta\right)}{N_{\exp}}. \tag{129}$$

*Once again, we assume the same conditions on $\mathcal{F}$ as required in Theorem 2.*

### B.3 Extension to unbounded discriminator families

Note that when the family of discriminators $\mathcal{F}$ does not have finite cardinality, it in fact suffices to just bound the imitation gap against a finite covering of $\mathcal{F}$. We spell out the details explicitly below.

In particular, we can replace $F_{\max}$ by $\max_{t \in [H]} \mathcal{N}(\mathcal{F}_t, 1/N_{\exp}, \|\cdot\|_\infty)$, where $\mathcal{N}(\mathcal{G}, \xi, \|\cdot\|)$ denotes the covering number of $\mathcal{G}$ in the norm $\|\cdot\|$ as defined below.

**Definition 5** (Covering number). *For a function class $\mathcal{G}$, tolerance $\xi$ and norm $\|\cdot\|$, the covering number $\mathcal{N}(\mathcal{G}, \xi, \|\cdot\|)$ is defined as the cardinality of the smallest set of functions $\mathcal{G}^\xi$ such that for each $g \in \mathcal{G}$, there exists a $g' \in \mathcal{G}^\xi$,*

$$\|g - g'\| \leq \xi. \tag{130}$$

**Corollary 1.** *When $\mathcal{G}$ is chosen as the set of $1$-bounded linear functions, $\mathcal{G} = \{\{\langle x, \theta\rangle : x \in \mathbb{B}_2^d\} : \theta \in \mathbb{B}_2^d\}$, where $\mathbb{B}_2^d$ denotes the $L_2$ unit ball in $\mathbb{R}^d$, $\mathcal{N}(\mathcal{G}, \xi, \|\cdot\|_\infty) \leq \left(\frac{2\sqrt{d}}{\xi} + 1\right)^d$.*

*Proof.* For any $g, g' \in \mathcal{G}$, where $g$ and $g'$ correspond to parameters $\theta, \theta' \in \mathbb{B}_2^d$,

$$\|g - g\|_\infty \leq \max_{x \in \mathcal{X}} \langle x, \theta - \theta'\} \tag{131}$$

$$\leq \|x\|_2 \|\theta - \theta'\|_2 \tag{132}$$

$$\leq \|\theta - \theta'\|_2. \tag{133}$$

Since the $L_2$ covering number of $\mathbb{B}_2^d$ is bounded by $\left(\frac{2\sqrt{d}}{\xi} + 1\right)^d$, the result immediately follows by defining the covering of $\mathcal{G}$ as $\{\langle\theta, \cdot\rangle : \theta \in \mathcal{K}\}$ where $\mathcal{K}$ is the optimal covering of $\mathbb{B}_2^d$ in $L_2$ norm. $\quad\square$

**Definition 6** (Discretization of discriminator space). *Define $\mathcal{F}_t^\xi$ as the optimal covering of $\mathcal{F}_t$ under the $L_\infty$ norm in the sense of Definition 5. The discretized family of discriminators we consider is, $\mathcal{F}^\xi = \otimes_{t=1}^H \mathcal{F}_t^\xi$.*

**Lemma 9.** *Suppose for all functions $f' \in \mathcal{F}_t^{\xi_1/H}$, simultaneously $J_{f'}(\pi^E) - J_{f'}(\pi^{RE}) \leq \xi_2$. Then, for all discriminators $f \in \mathcal{F}$, $J_f(\pi^E) - J_f(\pi^{RE}) \leq 2\xi_1 + \xi_2$.*

*Proof.* Consider any discriminator $f \in \mathcal{F}$. By construction, there exists an $f' \in \mathcal{F}^{\xi_1/H}$ such that,

$$\|f - f'\|_\infty \leq \xi_1/H. \tag{134}$$

Since for any policy $\pi$, the value $J_f(\pi)$ under a discriminator $f \in \mathcal{F}$ is an $H$-Lipschitz function of $f$, we can make a statement about how well $J_{f'}(\pi)$ approximates $J_f(\pi)$ for an appropriately chosen $f' \in \mathcal{F}^{\xi_1/H}$. In particular, the nearest (in $L_\infty$ norm) $f' \in \mathcal{F}^{\xi_1/H}$ to $f \in \mathcal{F}$ satisfies that for any policy $\pi$,

$$|J_f(\pi) - J_{f'}(\pi)| \leq H \times \frac{\xi_1}{H}. \tag{135}$$

As a consequence, for any discriminator $f \in \mathcal{F}$,

$$J_f(\pi^E) - J_f(\pi^{RE}) \leq |J_f(\pi^E) - J_{f'}(\pi^E)| + J_{f'}(\pi^E) - J_{f'}(\pi^{RE}) + |J_{f'}(\pi^{RE}) - J_f(\pi^{RE})| \tag{136}$$
$$\leq \xi_1 + \xi_2 + \xi_1 = 2\xi_1 + \xi_2. \tag{137}$$

$\square$

In particular, this means that if we minimize $J_{f'}(\pi^E) - J_{f'}(\pi^{RE}) \leq \xi_2$ for all $f' \in \mathcal{F}^{1/N_{\exp}H}$, then we can ensure that for all $f \in \mathcal{F}$,

$$J_f(\pi^E) - J_f(\pi^{RE}) \leq \frac{2}{N_{\exp}} + \xi_2. \tag{138}$$

This implies the following theorem,

**Theorem 9.** *Consider the policy $\pi^{RE}$ returned by Algorithm 1 where $\mathcal{F}$ is instead chosen as $\mathcal{F}^{\frac{1}{HN_{\exp}}}$ (as defined in Definition 6). Assume that $\pi^E \in \Pi$, the ground truth reward function $r_t \in \mathcal{F}_t$ which is assumed to be bounded (For all $f_t \in \mathcal{F}_t$, $\|f_t\|_\infty \leq 1$). Choose $|D_1|, |D_2| = \Theta(N_{\exp})$ and suppose $N_{\text{replay}} \to \infty$. With probability $\geq 1 - 3\delta$,*

$$J(\pi^E) - J(\pi^{RE}) \lesssim \mathcal{L}_1 + \mathcal{L}_2 + \frac{\log(\mathcal{N}_{\max}H/\delta) + 1}{N_{\exp}} \tag{139}$$

*where $\mathcal{N}_{\max} \triangleq \max_{t \in [H]} \mathcal{N}(\mathcal{F}_t, 1/HN_{\exp}, \|\cdot\|_\infty)$ corresponds to the maximal covering number of the function classes $\mathcal{F}_t$, and,*

$$\mathcal{L}_1 \triangleq H^2 \, \mathbb{E}_{\pi^E}\left[\frac{\sum_{t=1}^H \mathcal{M}(s_t, t)\mathsf{TV}\left(\pi_t^E(\cdot|s_t), \pi_t^{BC}(\cdot|s_t)\right)}{H}\right], \tag{140}$$

$$\mathcal{L}_2 \triangleq H^{3/2}\sqrt{\frac{\log(\mathcal{N}_{\max}H/\delta)}{N_{\exp}} \frac{\sum_{t=1}^H \mathbb{E}_{\pi^E}\left[1 - \mathcal{M}(s_t, t)\right]}{H}}.$$

**Remark 7.** *Note that this line of reasoning can be extended to Theorem 3 and Theorem 8 to show that the same guarantees as eq. (12) and eq. (129) respectively hold, but with $F_{\max}$ replaced by $\mathcal{N}_{\max}$.*

### B.4 Bounds on RE in the linear expert setting

In this section, we provide an upper bound on the imitation gap of RE in the presence of linear function approximation, which is studied in Rajaraman et al. [2021]. This is a special case of the case of IL under parametric function approximation with Lipschitzness. To avoid any ambiguity, we formally define IL with linear function approximation, which is the case when $(i)$ the expert follows

an unknown linear classifier in a known set of features, and $(ii)$ the reward function admits a linear parameterization.

The goal is to show that there exists a simple choice of the membership oracle such that the imitation gap of the resulting algorithm grows as $H^{3/2}$ and decay in the size of the expert dataset as $1/N_{\mathrm{exp}}$ up to logarithmic factors, breaking the error compounding barrier and achieving the optimal dependency on these parameters. We first introduce the linear setting below.

**Assumption 8** (Linear-expert setting). *For each $(s, a, t)$ tuple, the learner is assumed to have a feature representation $\phi_t(s, a) \in \mathbb{R}^d$. For each time $t$, there exists an unknown vector $\theta_t^E \in \mathbb{R}^d$ such that $\forall s \in \mathcal{S}$,*

$$\pi_t^E(s) = \arg\max_{a \in \mathcal{A}} \langle \theta_t^E, \phi_t(s, a) \rangle. \tag{141}$$

*i.e., the expert policy is deterministic and realized by a linear classifier. We assume that $\theta_t^E \in \mathbb{S}^{d-1}$ without loss of generality.*

**Definition 7** (Policy induced by a linear classifier). *Consider a set of vectors $\theta = \{\theta_1, \cdots, \theta_H\}$ where each $\theta_t \in \mathbb{R}^d$. A policy $\pi^\theta$ is said to be induced by the set of linear classifiers defined by $\theta$ if for all $s \in \mathcal{S}$ and $t \in [H]$,*

$$\pi_t^\theta(s) = \arg\max_{a \in \mathcal{A}} \langle \theta_t, \phi_t(s, a) \rangle. \tag{142}$$

*By this definition, $\pi^E = \pi^{\theta^E}$.*

**Definition 8** (Linear reward setting). *Define $\mathcal{R}_{\mathrm{lin,t}}$ as the family of linear reward functions (defined at the single time-step $t$) which takes the form of an unknown linear function of a set of the features,*

$$\mathcal{R}_{\mathrm{lin,t}} = \left\{ \{ r_t(s, a) = \langle \omega, \phi_t(s, a) \rangle : s \in \mathcal{S}, a \in \mathcal{A} \} : \omega \in \mathbb{R}^d, \|\omega\|_2 \leq 1 \right\}. \tag{143}$$

*For the rewards to be 1-bounded, we assume the features satisfy $\|\phi_t(s, a)\|_2 \leq 1$. Define $\mathcal{R}_{\mathrm{lin}} = \otimes_{t=1}^H \mathcal{R}_{\mathrm{lin,t}}$. The linear reward setting assumes the true reward function of the MDP, $r \in \mathcal{R}_{\mathrm{lin}}$.*

**Remark 8.** *Note that our guarantees in Theorem 10 hold even if the set of features in the definition of $\mathcal{R}_{\mathrm{lin,t}}$ in Definition 8 differ from those used to define the expert classifier Assumption 8. Regardless, we assume that both sets of features are known to the learner.*

In the case of parametric function approximation with Lipschitzness, note that we assume both the weak margin condition (Assumption 6), as well as the existence of a linear classification oracle (Assumption 5). Below, in the linear expert case, we show a sufficient condition which implies both of these conditions. In particular, define the positive hemisphere with pole at $\theta$, i.e. $\{x : \mathbb{B}_2^d : \langle \theta, x \rangle \geq 0\}$ as $\mathbb{H}_\theta^d$. We abbreviate $\mathbb{H}_{\theta_t^E}^d$ as $\mathbb{H}_t^d$.

**Assumption 9** (Bounded density assumption). *For each time $t \in [H]$, state $s \in \mathcal{S}$, action $a \in \mathcal{A}$ and $\theta \in \Theta_t$, define $\overline{\phi}_t(s, a) = \phi_t(s, a_s^\theta) - \phi_t(s, a)$ where $a_s^\theta = \arg\max_{a' \in \mathcal{A}} \langle \theta, \phi_t(s, a') \rangle$. Consider the measure $\mathrm{Pr}_{\pi^E} \left( \exists a \neq a_{s_t}^\theta : \overline{\phi}_t(s_t, a) \in \cdot \right)$. Let $\overline{d}_t^E$ represent the Radon-Nikodym derivative of this measure against the uniform measure on $\mathbb{H}_t^{d-1}$. The bounded density assumption states that for each $t \in [H]$ there are constants $c_{\min} > 0$ and $c_{\max} < \infty$ such that for all $x \in \mathbb{H}_t^d$,*

$$c_{\min} \leq \overline{d}_t^E(x) \leq c_{\max}. \tag{144}$$

We now state the main result we prove for IL in the linear setting.

**Theorem 10.** *Under Assumptions 8 and 9, appropriately instantiating $\mathtt{RE}$ ensures that with probability $\geq 1 - \delta$,*

$$J(\pi^E) - J(\pi^{\mathtt{RE}}) \lesssim \sqrt{\frac{c_{\max}}{c_{\min}}} \frac{H^{3/2} d^{5/4} \log^{\frac{3}{2}}(N_{\mathrm{exp}} dH/\delta)}{N_{\mathrm{exp}}}.$$

The proof of this result follows by showing that under Assumption 9, both the weak margin condition (Assumption 6) is satisfied, and the classification oracle (Assumption 5) can be constructed. We begin by showing the former.

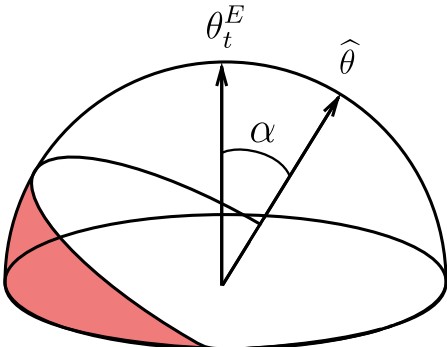

Figure 8: If at any time $t \in [H]$ and state $s$, $\phi_t(s, \pi_t^E(s)) - \phi_t(s, a)$ lies in the red shaded region for some action $a$, then, the action played by $\pi^{\text{BC}}$ and $\pi^E$ at this state are different.

**Lemma 10.** *Under Assumption 9, the $\alpha$-weak margin condition is satisfied with $\alpha = 1$ and $\mu = 2c_{\max}\sqrt{d}$. In particular, for all $\theta \in \mathbb{S}^{d-1}$,*

$$\Pr_{\pi^E}\left( \langle \theta, \phi_t(s, a_{s_t}^\theta) \rangle - \max_{a \neq a_{s_t}^\theta} \langle \theta, \phi_t(s, a) \rangle \geq \eta \right) \geq 1 - \left( 2c_{\max}\sqrt{d} \right)\eta. \tag{145}$$

*where $a_{s_t}^\theta \triangleq \arg\max_{a \in \mathcal{A}} \langle \theta, \phi_t(s, a) \rangle$.*

*Proof.* Observe that,

$$\Pr_{\pi^E}\left( \exists a \neq a_{s_t}^\theta : \langle \theta, \phi_t(s_t, a_{s_t}^\theta) \rangle - \langle \theta, \phi_t(s_t, a) \rangle \leq \eta \right) \tag{146}$$

$$= \Pr_{\pi^E}\left( \exists a \neq a_{s_t}^\theta : \phi_t(s_t, a_{s_t}^\theta) - \phi_t(s_t, a) \in \{ x \in \mathbb{H}_\theta^d : \langle x, \theta \rangle \leq \eta \} \right) \tag{147}$$

$$\stackrel{(i)}{=} \Pr_{\pi^E}\left( \exists a \neq a_{s_t}^\theta : \overline{\phi}_t(s_t, a) \in \{ x \in \mathbb{H}_\theta^d : \langle x, \theta \rangle \leq \eta \} \right) \tag{148}$$

$$\stackrel{(ii)}{\leq} c_{\max}\Pr(\langle U, \theta \rangle \leq \eta) \tag{149}$$

where in $(i)$, $\overline{\phi}_t$ is as defined in Assumption 9 and in $(ii)$, $U$ is uniformly distributed on the unit hemisphere, $\mathbb{H}_\theta^d$. Note that $(ii)$ follows from the bounded density condition, Assumption 9. Note that the RHS essentially corresponds to the volume (probability measure) of a disc of height $\eta$ cut out of a sphere from the center. Up to normalization factors, this can be upper bounded by the surface area of the base of the disc, multiplied by the height of the disc. Namely,

$$\frac{\eta \times \frac{\pi^{\frac{d-1}{2}}}{\Gamma\left(\frac{d-1}{2}+1\right)}}{\frac{1}{2}\frac{\pi^{\frac{d}{2}}}{\Gamma\left(\frac{d}{2}+1\right)}} \tag{150}$$

Using Gautschi's inequality, for any $x \geq 0$ and $\ell \in (0, 1)$ $x^{1-\ell} \leq \frac{\Gamma(x+1)}{\Gamma(x+\ell)} \leq (1+x)^{1-\ell}$. With $\ell = \frac{1}{2}$, $\frac{\Gamma(\frac{d}{2}+1)}{\Gamma(\frac{d+1}{2})} \leq \sqrt{d}$. Combining with eq. (149) results in,

$$\Pr_{\pi^E}\left( \exists a \neq a_{s_t}^\theta : \langle \theta, \phi_t(s_t, a_{s_t}^\theta) \rangle - \langle \theta, \phi_t(s_t, a) \rangle \leq \eta \right) \leq 2c_{\max}\sqrt{d}\eta \tag{151}$$

Therefore the probability of the complement event is lower bounded by $1 - 2c_{\max}\sqrt{d}\eta$, completing the proof. $\square$

The final thing to show is that the bounded density assumption can also be used to construct a classification oracle in the sense of Assumption 5.

In particular, as discussed in the main paper, we show that algorithms for minimizing the generalization error, can be used to construct a classification oracle. The compression based algorithm of

Daniely and Shalev-Shwartz [2014] provides a guarantee on the generalization error. From Theorem 5 of Daniely and Shalev-Shwartz [2014], in the realizable setting, for linear classification, the resulting classifier $\hat{\theta}$ has expected 0-1 loss upper bounded by $(d + \log(1/\delta)) \log(n)/n$, given $n$ classification examples. Namely, in the notation of Assumption 5, the resulting classifier $\hat{\theta}$ satisfies with probability $\geq 1 - \delta$,

$$\Pr_{s \sim \mathcal{D}} \left( \arg\max_{a \in \mathcal{A}} f_{\theta^*}(s^i, a) \neq \arg\max_{a \in \mathcal{A}} f_{\hat{\theta}}(s^i, a) \right) \leq \frac{(d + \log(1/\delta)) \log(n)}{n}. \tag{152}$$

Next we show that under Assumption 9, this equation can be used to bound the error in the parameter space, $\|\theta^* - \hat{\theta}\|_2$. Namely, in Assumption 5, we may choose $\mathcal{E}_{\mathbb{B}_2^d, n, \delta}$ as $\asymp \frac{(d + \log(1/\delta)) \log(n)}{n}$, up to constants depending on $c_{\min}$.

**Lemma 11.** *Consider the compression based learner $\widehat{\theta}_t^{BC} = \hat{\theta}_t$ of Daniely and Shalev-Shwartz [2014] for multi-class linear classification. Then, under Assumption 9, with probability $\geq 1 - \delta$,*

$$\|\widehat{\theta}_t^{BC} - \theta_t^*\|_2 \leq \frac{2\pi}{c_{\min}} \frac{(d + \log(1/\delta)) \log(N_{\exp})}{N_{\exp}} \tag{153}$$

*Proof.* Fix $t \in [H]$. The generalization error of $\widehat{\theta}_t^{BC} = \hat{\theta}_t$ can be written as,

$$\Pr_{\pi^E} \left( \arg\max_{a \in \mathcal{A}} \langle \theta_t^*, \phi_t(s_t, a) \rangle \neq \arg\max_{a \in \mathcal{A}} \langle \widehat{\theta}_t^{BC}, \phi_t(s_t, a) \rangle \right)$$
$$= \Pr_{\pi^E} \left( \exists a \neq \pi_t^E(s_t) : \phi_t(s_t, \pi_t^E(s_t)) - \phi_t(s_t, a) \in \mathcal{C} \right), \tag{154}$$

where $\mathcal{C}$ is illustrated in fig. 8 and is formally defined as,

$$\mathcal{C} \triangleq \{x \in \mathbb{H}_t^d : \langle x, \widehat{\theta}_t^{BC} \rangle \leq 0\}. \tag{155}$$

On the states which "belong" to $\mathcal{C}$ (i.e. at those states $s$ where $\exists a \neq \pi_t^E(s_t) : \phi_t(s, \pi_t^E(s_t)) - \phi_t(s, a) \in \mathcal{C}$), there exists an action $a$ such that $\widehat{\theta}_t^{BC}$ is better correlated with this action than $a_s^*$. In other words, $\widehat{\theta}_t^{BC}$ and $\theta^*$ play different actions at this state. Note that $\mathcal{C}$ is essentially the set difference of two hemispheres with different poles. By the bounded density condition, Assumption 9, and eq. (154),

$$\Pr_{\pi^E} \left( \pi_t^E(s_t) \neq \arg\max_{a \in \mathcal{A}} \langle \widehat{\theta}_t^{BC}, \phi(s, a) \rangle \right) \geq c_{\min} \Pr(U \in \mathcal{C}), \tag{156}$$

where $U$ is uniformly distributed over $\mathbb{H}_\theta^d$. Referring to fig. 8, we have that,

$$\Pr(U \in \mathcal{C}) = \frac{\alpha}{\pi} \tag{157}$$

where $\alpha$ is the angle between $\widehat{\theta}_t^{BC}$ and $\theta_t^E$. In particular, from eq. (156),

$$\Pr_{\pi^E} \left( a_s^* \neq \arg\max_{a \in \mathcal{A}} \langle \widehat{\theta}_t^{BC}, \phi(s, a) \rangle \right) \geq c_{\min} \frac{\alpha}{\pi} \geq c_{\min} \frac{\|\theta^* - \widehat{\theta}_t^{BC}\|_2}{\pi}, \tag{158}$$

where in the last inequality, we use the fact that $\|\theta^*\|_2 = \|\widehat{\theta}_t^{BC}\|_2 = 1$ without loss of generality. By the generalization error bound on $\widehat{\theta}_t^{BC} = \hat{\theta}_t$ in eq. (152), with probability $\geq 1 - \delta$,

$$\|\theta^* - \widehat{\theta}_t^{BC}\|_2 \leq \frac{\pi}{c_{\min}} \frac{(d + \log(1/\delta)) \log(N_{\exp})}{N_{\exp}} \tag{159}$$

$\square$

Lemma 11 shows that under the bounded density condition Assumption 9, the compression based learner $\hat{\theta}$ of Daniely and Shalev-Shwartz [2014] essentially induces a classification oracle for linear classification with $\mathcal{E}_{\mathbb{B}_2^d, n, \delta} = \frac{\pi}{c_{\min}} \frac{(d + \log(1/\delta)) \log(n)}{n}$. Finally, from Corollary 1, we have a bound on the covering number of linear families. Putting together all of these results with Theorem 3 (noting that we can replace $F_{\max}$ by $\mathcal{N}_{\max}$ from Remark 7) results in Theorem 10.

# C Additional experimental results

We include some additional experimental results in this section of the paper.

In fig. 9, we plot the distributions of the prefix weights generated by each membership oracle on simulated BC rollouts on WalkerBulletEnv. Note that $\mathcal{M}_{\text{VAR}}$ is significantly overconfident in prefix weights compared to $\mathcal{M}_{\text{EXP}}$, as indicated by the heavier right-tail. On the other hand, $\mathcal{M}_{\text{RND}}$ and $\mathcal{M}_{\text{MAX}}$ are less overconfident and better overlap with the idealized prefix weights induced by $\mathcal{M}_{\text{EXP}}$. This aligns with the correlation plot between the various membership oracles in Fig. 5. Moreover, in terms of policy performance, this further justifies the superior behavior of $\mathcal{M}_{\text{MAX}}$ compared to $\mathcal{M}_{\text{VAR}}$.

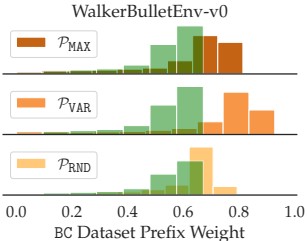

Figure 9: Histogram of prefix weights generated by rolling out trajectories from BC . The green superimposed histogram represents prefix weights generated by $\mathcal{M}_{\text{EXP}}$

In Fig. 10, we consider how each of the changes we described in the main section of the paper lead to improved performance of our RE baseline. The first, using a Wasserstein distance, leads to lower expected return but is required for solving the full moment-matching problem – see Swamy et al. [2021] for more details. Switching from PPO to the more sample-efficient SAC [Haarnoja et al., 2018] leads to fast learning. Adding in gradient penalties for discriminator stability [Swamy et al., 2021, Gulrajani et al., 2017] also improves final performance and learning speed. The last change we employ, using Optimistic Mirror Descent [Daskalakis et al., 2017] in both the discriminator and RL algorithm also (slightly) improves performance. To our knowledge, we are the first to utilize this technique in the imitation learning literature and reccomend it as best practice for future moment-matching algorithms. We refer interested readers to the work of Syrgkanis et al. [2015] for theoretical details of why OMD enables superior performance.

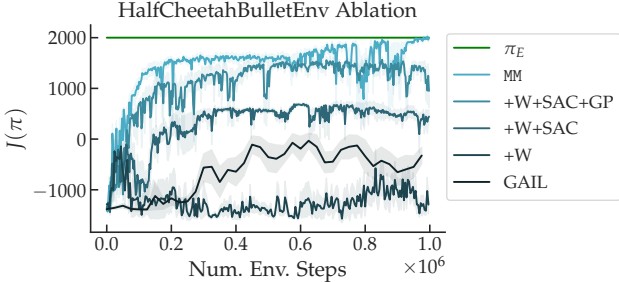

Figure 10: We ablate the four key changes we made to off-the-shelf GAIL to improve performance / theoretical guarantees. We see that each improved performance, with MM significantly out-performing options with fewer changes. Our improvements upon MM with the Replay Estimation technique are therefore improving upon an already strong baseline.

# D    Experimental Setup

We begin with the hyperparameters for our Standard Bullet and Noisy Bullet experiments.

## D.1    Expert

We use the Stable Baselines 3 [Raffin et al., 2021] implementation of PPO [Schulman et al., 2017] or SAC [Haarnoja et al., 2018] to train experts for each environment. For the most part, we use already tuned hyperparameters from [Raffin, 2020] in the implementation. The modifications we used are are shown in table 1.

| PARAMETER | VALUE |
|---|---|
| BUFFER SIZE | 300000 |
| BATCH SIZE | 256 |
| $\gamma$ | 0.98 |
| $\tau$ | 0.02 |
| TRAINING FREQ. | 64 |
| GRADIENT STEPS | 64 |
| LEARNING RATE | LIN. SCHED. 7.3E-4 |
| POLICY ARCHITECTURE | 256 X 2 |
| STATE-DEPENDENT EXPLORATION | TRUE |
| TRAINING TIMESTEPS | 1E6 |

Table 1:  Expert hyperparameters for Walker Bullet Task and Hopper Bullet Task

### D.1.1    Noisy Experts

In addition to the default Bullet Tasks, we test performance of algorithms on noisy environments. Namely, we generate noisy expert data by re-training expert policies with Gaussian noise added to the actions of the expert during the exploration phase while training. We then re-generate expert data by sampling from the expert policies trained on noisy data to analyze the performance of our method under stochasticity. Table 2 lists the standard deviation of the (i.i.d.) noise we applied to the actions in the different environments.

| ENV. | NOISE DISTRIBUTION. |
|---|---|
| HOPPER | $\mathcal{N}(0, 0.1)$ |
| WALKER | $\mathcal{N}(0, 0.5)$ |

Table 2:  Noise we applied to all policies in each environment.

## D.2    Baseline

We average over 5 runs and use a common architecture of 256 x 2 with ReLU activations for both our method and the MM baseline we compare against. For each datapoint, the cumulative reward is averaged over 10 trajectories. For all tasks, we train on $\{6, 12, 18\}$ expert trajectories with a maximum of 400k iterations of the optimization procedure. Table 3 shows the hyperparameters we used for MM. Empirically, smaller learning rates, large batch sizes, and gradient penalties were critical for the stable convergence of our method.

We note that MM requires careful tuning of $f$ UPDATE FREQ. for strong performance. We searched over step sizes of $\{1250, 2500, 5000\}$ and selected the one which achieved the most stable updates. In practice, we recommend evaluating a trained policy on a validation set to set this parameter.

We also used similar parameters for training SAC, also from the Stable Baselines 3 [Raffin et al., 2021] implementation, as we did for training the expert policy. Table 4 shows the choice of hyperparmeters we used for training SAC. We directly added in the optimistic mirror descent optimizers [Daskalakis et al., 2017] for both the critic and actor objectives of SAC.

| PARAMETER | VALUE |
|---|---|
| BATCH SIZE | 2048* |
| LEARNING RATE | LINEAR SCHEDULE OF 8E-3* |
| $f$ UPDATE FREQ. | 5000 |
| $f$ GRADIENT TARGET | 0.4 |
| $f$ GRADIENT PENALTY WEIGHT | 10 |

Table 3: Learner hyperparameters for MM . * indicates the parameter was different for the Hopper Initial State shift experiments (4096 for batch size and Linear Schedule of 8e-4, respectively.).

| PARAMETER | VALUE |
|---|---|
| $\gamma$ | 0.98 |
| $\tau$ | 0.02 |
| TRAINING FREQ. | 64 |
| GRADIENT STEPS | 64 |
| LEARNING RATE | LINEAR SCHEDULE OF 7.3E-4 |
| POLICY ARCHITECTURE | 256 X 2 |

Table 4: Leaning hyperparameters for the SAC component of MM

Table 5 shows the learning hyperparameters for any BC policies used for generating simulated data for the membership oracles. Table 6 shows the number of training steps per task we used for both the baseline and our method.

| PARAMETER | VALUE |
|---|---|
| ENTROPY WEIGHT | 0 |
| L2 WEIGHT | 0 |
| TRAINING TIMESTEPS | 1E5 |

Table 5: Learner hyperparameters for Behavioral Cloning

### D.3 Our Algorithm

In this section, we use **bold text** to highlight sensitive hyperparameters. We use the same network architecture choices as the MM baseline. For all environments, we generated 100 trajectories of simulated behavior cloning data to use with our method.

For all tasks, we rolled out **100 trajectories** from a BC trained network to use with our membership oracle. Table 7 shows how we partitioned our dataset between the BC training set and the expert membership oracle dataset. We also use the full dataset for moment matching, not just $D_2$, as we found this lead to slightly better performance.

### D.4 Membership Oracle Parameters

For both $\mathcal{M}_{\text{VAR}}$ and $\mathcal{M}_{\text{MAX}}$, we use 5 BC networks in the ensemble. We followed the exact same parameters in Table 5 to train each BC imitator. Table 8 shows the choice of $\mu$ and $\beta$ values we used for each membership oracle.

### D.5 Initial State Shift Experiments

For these experiments, we used demonstrations generated by an expert trained on the standard Bullet tasks but subject the learner (both at train and test time) to a initial velocity perturbation of a zero-mean Gaussian with variance ($\sigma = 1e - 7$). We refer interested readers to our code for our precise method of injecting noise as we believe it might be of interest for future experiments. We note that in all the demonstrations, we see the expert start from rest. Despite this relatively small

| ENV. | TRAINING STEPS |
|---|---|
| WALKER (NO NOISE) | 400000 |
| WALKER (WITH NOISE) | 400000 |
| HOPPER (NO NOISE) | 400000 |
| HOPPER (WITH NOISE) | 400000 |

Table 6: Number of training steps for the different tasks

| EXPERT SIZE | $D_1$ | $D_2$ |
|---|---|---|
| 6 TRAJS | 4 | 2 |
| 12 TRAJS | 10 | 2 |
| 18 TRAJS | 16 | 2 |

Table 7: Partition of trajectories into $D_1$ and $D_2$ based on the number of expert trajectories provided. For the Noisy Walker experiments, we used $5, 10, 14$ trajectories for $D_1$ instead of the above.

| ENV | PARAMETER | $\mathcal{M}_{EXP}$ | $\mathcal{M}_{\text{RND}}$ | $\mathcal{M}_{\text{VAR}}$ | $\mathcal{M}_{\text{MAX}}$ |
|---|---|---|---|---|---|
| WALKER | $\beta$ | 0.1 | 0.1 | 0.01 | 0.1 |
| WALKER | $\mu$ | 0.33 | 0.22 | 0.015 | 0.35 |
| HOPPER | $\beta$ | 0.8 | 0.25 | 0.08 | 0.1 |
| HOPPER | $\mu$ | 0.68 | 0.4 | 0.05 | 0.25 |

Table 8: Membership oracle hyperparameters across different environments

shift, we see BC performance drop significantly, as is characteristic of real-world problems where it significantly under-performs on-policy IL methods. All results are averaged over five seeds.

For all environments, we train BC for 1e5 steps (as well as for the query policies for RE ).

For RE , we train 5 policies and use the max-distance approximate membership oracle. We use the above parameters for MM for our base moment-matcher.

| ENV | PARAMETER | $\mathcal{M}_{\text{MAX}}$ |
|---|---|---|
| WALKER | $\beta$ | 0.01 |
| WALKER | $\mu$ | 0.0001 |
| HOPPER | $\beta$ | 0.01 |
| HOPPER | $\mu$ | 0.0001 |

Table 9: Membership oracle hyperparameters across different initial state shift environments.