# OpenReview forum: "Minimax Optimal Online Imitation Learning via Replay Estimation"
_NeurIPS.cc/2022/Conference — NeurIPS 2022 Accept_

### Official Review · Reviewer_hKQj · 2022-06-30

**Rating:** 7
**Confidence:** 4
**Soundness:** 2 fair
**Presentation:** 3 good
**Contribution:** 3 good

**Summary:**

This paper studies the problem of online imitation where the environment interaction is allowed. This paper focuses on the moment matching method. The vanilla method directly estimates the expert's state-action distribution and therefore suffers a sample barrier issue when the sample size is sufficiently large. To address this issue, this paper extends the idea presented in Rajaraman et al. (2020) with new algorithm designs and analyses. Specifically, the proposed algorithm in this paper can work with function approximation, whereas the MIMIC-MD algorithm in Rajaraman et al. (2020) is only applied to the tabular setting. In terms of the theoretical analysis, this paper shows an imitation gap $\widetilde{O}(H^{3/2} d^{5/4}/N_{\exp})$ in the linear function approximation setting, under weaker assumptions compared with that in Rajaraman et al. (2021a). Finally, experiments on tasks from the PyBullet suite show the efficacy of the proposed algorithm.

=====

After the rebuttal, most of my concerns are well addressed. I recommend accepting this paper for two reasons. First, it makes a significant step in improving the worst-case bound of MM-style algorithms. Second, it investigates an interesting experiment setup. The main criticism of this paper includes insufficient related work (refer to the Limitations section) and weak experiment results (refer to my last reply on the discussion page). Based on the assumption that the authors can fix all these issues before the final version, I recommend accepting this paper and increase my score from 5 to 7.



**Questions:**


- Can the authors help explain why MM has different algorithm behaviors in Figure 2 and Figure 3?

 This paper claims that MM might fail on the MDP shown in Figure 2, but later on, it says that it can take the green action (expert action) on the MDP shown in Figure 3. However, the theoretical guarantee in Theorem 3.1 for MM cannot tell me why MM is sometimes good and sometimes bad. I realize algorithm properties of MM are much studied in the prior work [1]. Are there new insights in this paper?

- Why is the performance of MM bad on Pybullet Env? Is the superior performance of RE really benefited from a better estimation?

I observe that the performance of MM is bad in Figure 5, and I am surprised by this result because I believe that MM should perform well. To support my thought, I conducted an experiment by running the DAC algorithm without any tunning (the experiment configuration follows the code given by the authors). DAC is a particular case of MM when the function class allows a KL-divergence to measure the state-action distribution matching. Then, I find that with 10 expert trajectories, DAC can achieve the performance of 1500 ($\pm 500$) on HopperBulletEnv and 2000 ($\pm 200$) on Walker2dBulletEnv. I believe further tuning can boost the performance.

The main concern is when can RE be better than MM? MM-style algorithms are known to perform well on many tasks, even though it may suffer a sample barrier issue in the worst case. The authors are recommended for the discussion in Section 3.3 of [1], in which it is shown that RE has no clear advantage over MM in instances similar to locomotion control.

**Limitations:**

I appreciate and like the theoretical results developed in this paper. However, I still have some concerns.

**Important references are missing**

I believe the current references are somehow biased. I would suggest that the authors add a section of related work, if the authors agree. For example, similar instances in Figure 2 and Figure 3 and associated upper bound and lower bound for MM have been studied in [1]. Furthermore, the idea of RE (i.e., learning a BC policy to rollout) is also presented in [1] (though [1] studied the tabular setting). I think the connection with [1] should be clarified. It would be great to give credits for prior studies of MM algorithms (in other papers, it is called adversarial imitation learning algorithms) [2, 3, 4]. Otherwise, readers cannot give a full picture of MM algorithms.



**Gaps Between Theory and Practice**

- It is not clear for what kind of instances RE is better.  Note that we cannot claim that an algorithm is better than another if the upper bound is better.  This is because upper bounds could be loose. Even for instances in Figure 2 and Figure 3 of this paper, I do not know whether RE can be better than BC or MM, since there are no experiments or theoretical guarantees. Compared with the lower bound instances for BC and MM, I know RE can show an advantage in the large sample regime. But I do not know whether these hard cases occur in practice (or at least, whether these hard instances are similar to tasks in the benchmark considered in this paper).
- The improvement of RE is in the large sample regime but the experiment is in a small sample regime. From Figure 4, it seems that MM is better in a large sample regime ($N_{\exp} > |S|H$). However, the improvement magnitude should be minor in this regime, as the imitation gap of MM is already smaller than $\mathcal{O}(\sqrt{H})$. To better see this, let us consider the locomotion control tasks, in which H=1000. Then, the imitation gap of MM is smaller than 31, meaning that the improvement of a better algorithm can not be larger than 31. However, in experiments, it is shown that RE can be significantly better than MM in a small sample regime.
- MM should have good performance. This paper shows that MM has bad performance on Walekr2dBulletEnv. Though I do not find a good reference from the literature, my experiments show that MM can perform well; refer to my comments in the Questions section.

I would suggest that this paper can be rephrased to emphasize the theoretical contributions, if the authors agree. The motivation in section 2 in terms of ''better than both worlds'' is not very sound to me (the authors are encouraged to explain more about this motivation). Instead, I believe that a study of minimax optimality is sufficient to motivate this paper.

**Minor Issues**

- A right bracket is missing in Line 15 of the abstract.

- The MM family is too large and it is not clear which algorithm is studied in the main paper. In Appendix, it seems that TV distance is considered. I believe this point should be clarified in the main paper.  It is recommended to give more explanations about MM (though I prefer the name of state-action distribution matching). Otherwise, it is hard to follow the discussion in Section 2.

- The sample regime in theoretical guarantees should be clarified. For example, the bound in Theorem 3.1 should be $\min \\{ H, H\sqrt{|\mathcal{S}|/N_{\exp}} \\}$. Otherwise, one may be confused that if $N_{\exp}= 1$, the theoretical guarantee in Theorem 3.1 is meaningless. In other words, the bound in Theorem 3.1 is only meaningful in a large sample regime where $N_{\exp} > |\mathcal{S}|$.

- In experiments, this paper claims that gradient penalty can improve the convergence rate. But, I do not realize this point. Instead, I know that the gradient penalty can help reduce the generalization error; see the discussion about the discriminator in [3-4].


I would like to recommend accepting this paper if the above concerns are well addressed. Basically, some misleading claims should be clarified.

**Strengths And Weaknesses:**

**Strengths**

- This paper introduces several new algorithm designs (soft membership and prefix weight). These components make the idea in Rajaraman et al. (2020)  more generable.
- This paper provides new and better theoretical guarantees for the proposed algorithm. These theoretical results are essential for studying the minimax optimality on a statistical side.

**Weakness**

- Some relevant references are missing. The discussion in Section 2 has been presented in the prior work; see [1]. The idea of learning a BC policy and rolling out it to collect more trajectories to improve the estimation is also considered in [1]. The theoretical analysis of MM (Theorem 3.1 and Theorem 3.2 in this paper) has been conducted in [1]. Prior studies [2-4] about MM (and variants) are not even mentioned.
- Gaps between theory and practice. This paper studies the worst-case bound of RE. But, it is not clear for which kind of instances and what kind of setting, RE can be better than other algorithms. Though this paper presents some numerical results to answer such questions, these results are not very convincing. Please refer to the detailed comments below.

[1] T. Xu, Z. Li, Y. Yu, and Z.-Q. Luo. On generalization of adversarial imitation learning and beyond. arXiv, 2106.10424, 2022.

[2] Z. Liu, Y. Zhang, Z. Fu, Z. Yang, and Z. Wang. Provably efficient generative adversarial imitation learning for online and offline setting with linear function approximation. arXiv, 2108.08765, 2021b.

[3] Xu, T., Li, Z., & Yu, Y. (2020). Error bounds of imitating policies and environments. *Advances in Neural Information Processing Systems*, *33*, 15737-15749.

[4] Y. Wang, T. Liu, Z. Yang, X. Li, Z. Wang, and T. Zhao. On computation and generalization of generative adversarial imitation learning. In Proceedings of the 8th International Conference on Learning Representations, 2020.

---

> ### Author Response · Authors · 2022-08-02
> **Re:**
>
> We appreciate the reviewer’s enthusiasm about our work and would be happy to respond to the concerns raised.
>
> Weakness 1: We would be happy to cite [1] and add discussion of where our results overlap.
>
> Question 1: Sure. In Figure 2, because MM is trying to match the expert’s state distribution, it might take an action that deviates from what it saw in the demonstration (the red self-loop). In contrast, in Figure 3, the expert never visits s_x so MM has no incentive to take an action that leads to it.
>
> Question 2: We note that the experiments we conducted in this paper are on modified versions of standard PyBullet environments, in which we added noise to the dynamics or a shift to the initial state. We did this because, as observed by Swamy et al. and Spencer et al., behavioral cloning is able to match expert performance on the standard environments, which is contrast to most empirical observations. We would therefore appreciate clarification as to whether the reviewer ran experiments using these Gym wrappers and the demonstrations provided in our supplementary zip. Regardless, we believe the MM algorithm we use is quite a strong baseline – we conduct an ablation in a standard environment in Fig. 10 of the appendix and find that it significantly outperforms many other approaches. It might indeed be possible that approaches like DAC are better fits for the modified environments we use in this paper. However, one might still be able to use the replay estimation technique to improve their finite-sample performance. Put differently, the replay estimation technique is independent of the particular moment-matching algorithm used, which should be chosen in a problem-specific manner.
>
>
> Gaps 1: The reviewer is correct that it is quite difficult to, by comparing upper bounds, determine which algorithm will perform better on a particular problem. In general however, we would expect MM/RE to do better than BC on problems where there is a large amount of covariate shift between the learner and expert’s state visitation distribution, a common feature of real-world problems like autonomous driving. We would expect RE to do better than MM on problems where there exists a decent mass of states where the expert’s actions are low variance, enabling the repeated BC rollouts to be useful in nailing down these actions. For example, in the driving domain, the variance of actions on long, straight, highway lanes is going to be quite low compared to that of actors on merging ramps. The RE technique allows us to take advantage of this fact. For RE to be better than both MM and BC in practice, we would need both of these conditions to be true.
>
> Gaps 2: Because the PyBullet tasks are periodic (e.g. walking, https://www.youtube.com/watch?v=_6qWoDCPde0), the effective horizon of the task is much smaller (on the order of 20 timesteps or so).
>
> Better than Both Worlds: A natural suggestion might be to run both BC and MM for a given problem and pick whichever performs better. We wanted to ensure that our algorithm could, for some problems, perform better than this naive strategy.
>
> Minor Issue 1: Good catch, we will fix this.
>
> Minor Issue 2: Matching state-action distributions in TV distance is sufficient for matching expert performance. However, as argued by Swamy et al., the weaker (and necessary) condition is to match expert behavior on all the basis elements of the class of rewards functions. If one chooses this basis to be the indicator function basis, they recover TV distance, which is what we do for convenience in theory. However, all of our results still hold over more restricted classes, so long as they contain the true reward function.
>
> Minor Issue 3: We would be happy to clarify the sample regime.
>
> Minor Issue 4: We believe that gradient penalties being used to speed up convergence to Nash equilibrium is a well known technique in the WGAN [1] and IL [2] communities. We also see this behavior empirically in the ablation we conduct in the appendix in Figure 10.
>
>
> [1] https://arxiv.org/abs/1704.00028
>
> [2] https://arxiv.org/abs/2006.13258

---

> > ### Comment · Reviewer_hKQj · 2022-08-03
> > **Updated Comments**
> >
> > Thanks for the detailed response. Some of my concerns are well-addressed. To further improve the quality of this paper, I would like to update my comments.
> >
> > Weakness 1: I believe both [1] and this paper explored a particular MM algorithm (with TV distance). [1] mainly focused on explaining why MM may perform well in a small sample regime (for some tabular instances), but this paper focuses on the worst-case performance and the algorithm that can achieve the lower bound. There is no overlap in the main results, but clarifying the mentioned parts about MM and algorithm designs is great.
> >
> > Question 1: Thanks for the explanation. I can understand this point, but I am not sure whether general readers can easily follow the discussion. In my opinion, the confusing part is that MM has different optimal solutions in two cases, but a worst-case bound (mentioned in Line 91) cannot explain this. As Reviewer kTcU has a similar concern about Section 2, I suggest further improving the readability in Section 2. Maybe, more discussion about Figure 4 can replace that in Section 2.
> >
> > Question 2: I use the expert dataset in the supplementary zip, but I do not use the env wrapper. I am sorry that I miss the reset function in the OffsetWrapper, which should be critical. I totally accept this modification. I agree that existing benchmarks (due to the limited states and deterministic transitions) are not sufficient to fully disclose algorithm properties, and the testing tasks in this paper are indeed good. Please involve a discussion about this point if the page space allows. I believe this point is important for future research in the imitation learning community.
> >
> > However, some questions about the improvement still hold. The ablation study in the Appendix helps answer this question in some sense. But, a gap still exists due to the exploration issue in the online setting. Specifically, optimization, generalization, and exploration are correlated in the online setting, so the empirical results deserve further investigation. Anyhow, this issue is minor, and this part should be left for future work.
> >
> > Gaps 1: Your clarification is nice and easy to follow. Please consider adding this to the main text.
> >
> > Gaps 2: I observe that the maximum episode steps for PyBullet tasks are 1000 (also mentioned in Line 348). Do you mean that some steps are correlated, and the effective decision steps are only 20?
> >
> > Better than Both worlds:  Thanks for your explanation. I have a better sense now. However, I still believe "better than both worlds" is not the main motivation of this paper. If my understanding is correct, the theoretical results in this paper do not suggest that RE can be better than this naive strategy. I believe this paper makes a breakthrough in addressing the hardness of the online imitation learning problem instances. That is, for any algorithm designed for this class of instances, the adversary cannot pick up a "harder" instance such that the most intelligent algorithm (actually RE) suffer more than $O(H^{3/2}/\epsilon)$ sample complexity. This differs from your explanation about "better than both worlds", in which an instance was given in advance. Please correct me if I am wrong.
> >
> > Minor Issue 2: I am not sure whether the lower bound instance can easily be extended to other cases. I believe the proof in Appendix somehow relies on the TV distance property. If you have alternative solutions, I am happy to see them. Otherwise, this issue is relatively minor, and clarification in the main text is fine.
> >
> > Minor Issue 4: I know such empirical results, and let me further clarify my concern: I do not realize whether the gradient penalty can improve something like the condition number to accelerate the convergence. The reported empirical results could be benefited from both optimization (i.e., convergence rate) and generalization. For example, I know that for MM with KL-divergence, without gradient penalty, the problem is ill-conditioned in the tabular case (because some non-visited states incur an infinite loss). In this case, the gradient penalty is required to ensure that the discriminator cannot have strong discriminative ability. Also,  the generalization bound (and practical performance) can be improved in this case. I mention my concern in the review, because I just want to check whether I miss some messages that the gradient penalty can also help from the viewpoint of optimization theory.
> >
> > Finally, do you have a plan to update a revised version of your paper? If so, please consider the above suggestions, and I will be happy to read the new paper.

---

> > > ### Author Response · Authors · 2022-08-05
> > > **Re:**
> > >
> > > We appreciate the quick response. Overall, we’d be happy to make the changes you suggested and appreciate your feedback in terms of finding areas of the current draft that are confusing to read.
> > >
> > > 1. We’ll clean up the presentation of Section 2 and include more discussion around Figure 4 and the issues facing Empirical Moment-Matching (drawing upon the more explicit discussion we presented above).
> > > 2. We’d be happy to add more discussion in the experiments section on the changes we made to the standard environments: emphasizing why the experiments we do are moving us towards more realistic scenarios in IL research would definitely make the paper stronger.
> > > 3. Thanks! We’d be happy to add our above explanation for Gap 1 to the main paper.
> > > 4. Yup! While the tasks are literally H=1000, the optimal policies (and expert demonstrations) have periodic behavior with H~=20 (see the attached video). This means that if the learner is able to get 20-step motion correct, they have a good chance of performing well for the entire horizon. In addition, we note that our bounds hold for the worst case instances, which for H=1000 can be far more pathological than ones with this sort of periodic structure.
> > > 5. Your point about “better than both worlds” is correct. Here, we intended the term to be used as: “using a careful combination of two canonical algorithms, BC and MM, one can achieve worst case performance which is better than what is achievable by either algorithm”. In a sense, the RE algorithm is better than both worlds, as it is able to combine the benefits of both algorithms. In the multi-armed bandits literature, the term better-than-both-worlds is used to mean what you suggested. In our paper, we *do not* intend the same usage of this term.
> > > 6. The lower bound result does not easily carry over when the function class is more restricted. However, the upper bound does. In the worst case, the reward function is exactly known (i.e. the discriminator function class has only 1 function in it) and the problem can be solved efficiently using dynamic programming without any expert trajectories, when the transition is known. Designing lower bounds for more restricted function classes therefore must depend carefully on the discriminator class considered.
> > > 7. Got it. We didn’t intend to make any optimization-theoretic claims about the effects of gradient penalties on speeding up convergence to Nash equilibria. We simply wanted to mention that in practice, when one uses deep networks as their discriminator class, gradient penalties have repeatedly been shown to improve empirical convergence rate. We’d be happy to add more discussion around this point.
> > >
> > > Overall, there are few different additions you’ve suggested (e.g. to Section 2, the experiments section, and some parts of the theory section). We definitely agree that these additions would make the paper stronger and improve its clarity. We would be more than happy to make these changes, but feel that it would be hard to make all of these modifications within the current 9 page limit (which, as far as we can tell, is not extended during this period https://neurips.cc/Conferences/2022/PaperInformation/NeurIPS-FAQ#:~:text=What%20is%20the%20page%20limit,papers%20for%20the%20camera%2Dready). If it is alright with you, we’d be happy to make them in the camera-ready version subject to the longer 10 page limit.

---

> > > > ### Comment · Reviewer_hKQj · 2022-08-06
> > > > **Thanks for Clarification**
> > > >
> > > > I really appreciate the authors' clarification. Thanks a lot! Most of my concerns are addressed, and I am ready to adjust my recommendation. Before that, I notice the comment from Reviewer Jnmr and want to discuss the experiment results.
> > > >
> > > > 1. I agree with the thought of Reviewer Jnmr regarding the horizon. In fact, the episode ends if the robot repeats a wrong action for at most 5-15 steps in the MuJoCo locomotion control tasks. Thus, the effective horizon may not be exactly 1000, it is definitively larger than 20.
> > > >
> > > > 2. Is the expert policy deterministic in the Standard Pybullet tasks (e.g., Walker2DBulletEnv in Figure 5)?
> > > >
> > > > 3. If my understanding is correct, there is no initial state shift for the Walker2dBulletEnv in Figure 5 (left). If so, my experiments about DAC are fair, and MM could have a strong performance. Of course, I agree that RE is independent of the implementation of a particular MM algorithm, and proper implementation of RE can boost the performance in the worst case.
> > > >
> > > > 4. For tasks with stochastic experts (the left two columns of Figure 6), both BC and RE can perform well, but the performance of MM is poor if $N$ (the expert sample size) is small. I have an explanation for this result. The sample complexity of MM depends on the action space size, if the expert policy is stochastic. Instead, the expert sample complexity of BC is regardless of the action space size, despite the randomness in the expert policy (an important message in (Rajaraman et al. (2020)). Since RE builds on BC, it is expected that it can perform well in the stochastic expert setting. Of course, the current theory of RE relies on the deterministic expert assumption, and I have no rigorous theoretical support for my explanation.
> > > >
> > > > 5. From the right two columns of Figure 6, there is no sufficient evidence showing that RE could be better than MM. With the initial state shift, it is reasonable to see that the performance of BC is poor. Furthermore, for the experiment in the most right column of Figure 6, the performance of MM becomes worse when $N$ increases. Could you help explain this strange phenomenon? (I have no good ideas, and I guess it is due to the implementation of membership oracle)
> > > >
> > > > 6. I observe that the ablation study in Figure 10 in Appendix D is based on the HalfCheetahEnv rather than the HopperEnv and Walker2dEnv. However, most empirical comparisons are based on the HopperEnv and Walker2dEnv. Could you help explain this mismatch?
> > > >
> > > > 7. The last message I want to mention is that if we test RE on the offline lower bound instance in (Rajaraman et al. (2020), its performance should be no better than MM's. This observation is already pointed out in the prior work [1]. Nevertheless, experiments in this paper tell us more about the performance of BC, MM, and RE in all kinds of tasks, which are valuable.

---

> > > > > ### Author Response · Authors · 2022-08-06
> > > > > **Re:**
> > > > >
> > > > > 1. We agree that the horizon might not be exactly 20. We mostly wanted to provide a concrete reason for why the worst-case bounds we derived might not apply to the experiments we ran.
> > > > > 2. The demonstrations we trained on were generated by deterministic expert policies.
> > > > > 3. Yes, for Fig. 5 (left) only, we do not modify the original problem. We agree with your points that "RE is independent of the implementation of a particular MM algorithm" and that a "proper implementation of RE can boost the performance in the worst case."
> > > > > 4. We use deterministic expert policies for all demonstrations (as is required for the MIMIC-MD / RE theory to hold).
> > > > > 5. We agree that MM/RE perform comparably on the Walker experiment (both outperforming BC). For the Hopper experiment, we performed limited tuning of the parameters for the membership oracle -- more careful tuning here might be able to lead to uniformly high performance. Regardless, we see RE out-perform BC, which is what we were hoping to demonstrate with this experiment.
> > > > > 6. The left half of figure 6 shows that on noisier (harder) versions of Walker/Hopper, MM is able to perform reasonably well with larger amounts of data. We chose to perform an ablation on HalfCheetah to show that this wasn't true for just these 2 environments.
> > > > > 7. We appreciate that you found our experiments valuable.

---

> > > > > > ### Comment · Reviewer_hKQj · 2022-08-07
> > > > > > **Could You Explain The Noisy Experts Setting?**
> > > > > >
> > > > > > Thanks for your response. I am sorry that I am confused about the noisy expert setting. I read Appendix E.1.1 but do not fully understand this setup. Do noisy experts mean that there is a Gaussian noise for the expert policy? Or does it mean that the environment transition is stochastic by injecting the Gaussian noise?

---

> > > > > > > ### Author Response · Authors · 2022-08-07
> > > > > > > **Re:**
> > > > > > >
> > > > > > > Sorry, this is a tricky point: the expert is deterministic but we add Gaussian noise to their actions (and to those of the learner) before executing them in the environment. Given the learner observes the original, noiseless actions in the collected demonstrations, this can be thought of as the expert being deterministic but the environment being stochastic.

---

> > > > > > > > ### Comment · Reviewer_hKQj · 2022-08-08
> > > > > > > > **Suggestions on Experiments**
> > > > > > > >
> > > > > > > > Thanks for the clarification. I appreciate the experiment setup in this paper. But, I believe current empirical results are not sufficient to support some claims. Here are my suggestions for experiments.
> > > > > > > >
> > > > > > > > - Investigate other implementation choices of MM (e.g., DAC). Be careful about the empirical conclusions on MM.
> > > > > > > > - Consider advanced exploration strategies (e.g., initializing MM with BC policies), which may improve the performance of MM on hard tasks in Figure 6 (left). I do believe that even for stochastic transition environments, MM can still outperform BC on tasks with large covariate shifts (refer to the prior work [1]).
> > > > > > > > - Consider the absorbing state wrapper introduced in DAC's paper, which is crucial for practical performance. Currently, I do not find this part in the given codebase. My implementation of DAC has this wrapper. In particular, the lack of this wrapper may explain your observation that MM could fail on HalfCheetah. For completeness, please also consider providing the ablation study on Hopper or Walker2d.
> > > > > > > > - Consider the state normalization. I find that there is no state normalization in your codebase. Typically, this preprocessing can boost the empirical performance (by improving the condition number).
> > > > > > > >
> > > > > > > > For more tricks, please refer to the following reference. But, I believe that the mentioned two techniques (absorbing state wrapper and state normalization) are the most important.  Let us be cautious when drawing both theoretical and empirical conclusions.
> > > > > > > >
> > > > > > > > Orsini, Manu, et al. "What matters for adversarial imitation learning?." Advances in Neural Information Processing Systems 34 (2021): 14656-14668.
> > > > > > > >
> > > > > > > > =====
> > > > > > > >
> > > > > > > > My other concerns are addressed. Though current experiments have some flaws (at least in my opinion), I recommend accepting this paper and believe the authors can fix all mentioned issues in the final version.

---

### Official Review · Reviewer_Jnmr · 2022-07-11

**Rating:** 7
**Confidence:** 4
**Soundness:** 2 fair
**Presentation:** 3 good
**Contribution:** 2 fair

**Summary:**

This paper studies the imitation learning problem with general function approximation. The authors utilized the replay estimation (RE) technique to improve the moment matching (MM) method. To analyze RE with general function approximation, they first introduced the membership oracle M and proved a general performance guarantee as a function of M. Then they instantiated the membership oracle with an offline classification oracle and proved a meta-theorem under some assumptions on this offline classification oracle. Based on this result, they argued that when the parameter error matches the generalization error (i.e., $\mathcal{E} _ {\Theta, n, \delta} \asymp \mathcal{E}_{\Theta, n, \delta}^{\text {class}}$), the imitation gap of RE is better than that of MM. Finally, they showed the empirical performance of RE on some tasks.

**Questions:**

1. In lines 56-57, the authors argued that they achieved the best-known imitation gap under significantly weaker assumptions compared with the prior work [Rajaraman et al. (2021a)]. In my opinion, these two works impose two different types of assumptions and it is hard to say which one is weaker. Can the authors elaborate more on this point?

2. In eq.(6), it seems to be $P (s _ { 1 \cdots t-1 }) \rightarrow P (s _ { 1 \cdots t })$ since we should also consider the weight on $s_t$ for estimating the moment in time step $t$?

3. The weak margin condition in assumption 3 is hard to understand. Can the authors explain more about the meaning of this condition and why this condition is required in the analysis?


**Ethics Review Area:**

["I don’t know"]

**Limitations:**

The authors discussed the limitations and potential negative societal impact of their work.

**Strengths And Weaknesses:**

Strengths:
1. This paper conducts a rigorous theoretical study for online imitation learning with general function approximation. To handle the challenge brought by function approximation, the author introduced the membership oracle, which is novel in imitation learning literature.
2. To instantiate the membership oracle, they proposed an offline classification oracle that can achieve a small statistical error for parameter estimation. With this offline classification oracle, they proved a performance guarantee for the resultant algorithm, which contributes new analysis.


Weakness:

1. The idea of replay estimation is not novel in imitation learning. In the tabular and known transition setting, [1] leverages the transition function to obtain a better estimation of the expert’s moments. Furthermore, [2] extends this idea to the setting where the transition function is unknown but environment interactions are allowed. In [2], they employ BC’s policy to interact with the environment and utilize the collected trajectories to establish a better estimation. This is exactly the replay estimation technique used in this paper. However, the discussion about the relation to the existing work [2] is missing in this work.

2. Due to the first point, the main contribution of this paper is that they applied the replay estimation technique with general function approximation. However, in the setting of general function approximation, the proposed algorithm and analysis require access to a strong offline classification oracle (assumption 2). This offline classification oracle is required to have a small parameter estimation error. Note that this parameter estimation error does not consider the data distribution, in stark contrast to the generalization error used in supervised learning. Therefore, this offline classification oracle asks a much stronger requirement than learning a classifier with a small generalization error in supervised learning. Besides, the RE method is proved to have a better imitation gap than MM only when the parameter error matches the generalization error. However, the key question of when the parameter error matches the generalization error is not answered in this paper.


3. For the experiment part, I have the following two concerns.

   a). In experiments, the classical moment matching method has poor performance while BC enjoys superior performance. This observation largely conflicts with the well-accepted empirical observation that moment matching based methods can match expert’s performance even with few expert trajectories on locomotion tasks [3, 4, 5]. Besides, in [6], they evaluated a similar MM approach named AdIRL on locomotion tasks from the same Py-Bullet suite. They showed that AdIRL can match expert’s performance across all environments, which conflicts with the experimental observation in this paper. This contradiction raises a doubt about whether the MM method is implemented correctly in experiments.

   b). Besides, as far as I am concerned, the performance gap established in this paper cannot explain the superior performance of RE in experiments. As the bound $\min (H^{3 / 2} / N_{\exp }, H / \sqrt{N_{\exp }} )$ suggests, RE has a better performance gap than MM in the large data regime $N_{\exp} \succsim H $. Nevertheless, in experiments, they considered the low data regime where $H \approx 1000$ and $N_{\exp} \leq 20$. Therefore, I believe that the developed theory cannot explain the experimental results in this paper.


4. Some theoretical results in this paper have been established in the existing work [2]. In particular, for moment matching, the performance gap upper bound and a tight lower bound (Theorems 3.1 and 3.2) have been proved in [2]; see Theorem 1 and Proposition 2 in [2] for details. However, the discussion on [2] is missing in this work.

References:

[1]. Rajaraman, N., Yang, L. F., Jiao, J., and Ramachandran, K. Toward the fundamental limits of imitation learning. arXiv preprint arXiv:2009.05990, 2020.

[2]. T. Xu, Z. Li, and Y. Yu. On generalization of adversarial imitation learning and beyond. arXiv, 2106.10424, 2021.

[3] J. Ho and S. Ermon, “Generative adversarial imitation learning,” in Advances in Neural Information Processing Systems 29, 2016, pp. 4565–4573.

[4] I. Kostrikov, K. K. Agrawal, D. Dwibedi, S. Levine, and J. Tompson, “Discriminator-actor-critic: Addressing sample inefficiency and reward bias in adversarial imitation learning,” in Proceedings of the 7th International Conference on Learning Representations, 2019.

[5] K. Brantley, W. Sun, and M. Henaff, “Disagreement-regularized imitation learning,” in Proceedings of the 8th International Conference on Learning Representations, 2020.

[6] G. Swamy, S. Choudhury, J. A. Bagnell, and S. Wu, “Of moments and matching: A game-theoretic framework for closing the imitation gap,” in Proceeding of the 38th International Conference on Machine Learning, 2021, pp.10 022–10032.

---

> ### Author Response · Authors · 2022-08-02
> **Re:**
>
> We appreciate the thorough discussion provided by the reviewer above. We would like to clarify our perspective on some of the concerns raised.
>
> Weakness 1: We would be happy to cite the paper [2] and add comprehensive discussions about where the results overlap. We agree that the replay estimation technique was known (under a different name) in the tabular setting [1]. However, our work addresses the setting of function approximation both in theory and practice which presents significant challenges, and has not been considered in prior work, including both [1] or [2].
>
> Weakness 2: We agree with the reviewer on the point that our assumption is stronger than the standard supervised learning one. We perform analysis in the setting of linear classification in Appendix B.3 to show that there are mild conditions under which the parameter and generalization errors match. In the classification setup, the linear setting already includes many cases such as kernel methods. The extension to more general classification settings is an excellent area for future work.
>
> Weakness 3: Note that it has recently been observed that properly tuning behavioral cloning can in fact perform quite well and achieve SOTA performance on all of the standard Mujoco (Spencer et al.) and PyBullet (Swamy et al.) environments, rendering comparisons on these environments somewhat vacuous. We therefore conduct experiments in more difficult versions of the standard environments (either by adding noise to the dynamics or adding an initial state shift), making it difficult to directly compare the performance in our experiments to that in prior work. We believe the MM baseline we use in experiments is quite strong – we include an ablation in Fig 10. of the Appendix where our method is able to out-perform other MM methods by a wide margin.
>
> Weakness 4: All of the standard PyBullet tasks are essentially periodic (https://www.youtube.com/watch?v=_6qWoDCPde0), rendering their effective horizon much smaller, on the order of H=20. This is the regime in which our theory also operates, where the number of samples observed is comparable to the number of episodes. However, note that in our theory, we focus on worst-case bounds, which can of course differ from the performance observed on specific instances.
>
>
> Q1. Our results generalize those of Rajaraman et al. out of the tabular setting. We are able to achieve a bound on the imitation gap that extends that in the prior work in the tabular setting to the case of general function approximation.
>
> Q2. Good catch, we will update this.
>
> Q3. The weak margin condition can be interpreted using the following example. Consider a neural network for classification into 3 classes - i.e. the network has 3 "heads" and the class returned is the argmax of the 3 head outputs. Then the margin condition states there is no classifier (i.e. choice of the network weights) such that a large probability mass of the population examples are "close" to a decision boundary. This means that the outputs of the 3 heads of the network are such that the largest head output (i.e. output of the predicted class) is larger than the remaining head outputs with some gap. In other words, if one were to perturb the 3 logits slightly, a large fraction of the examples should continue to remain in the same class as before. From this point of view, the assumption is reasonable and can be thought about as robustness of the “labels” to input perturbations.

---

> > ### Comment · Reviewer_Jnmr · 2022-08-06
> > **Response to the authors**
> >
> > Thank authors for the thoughtful response! Some of my concerns are addressed. Below are some remaining questions.
> >
> > Concern (b) in weakness 3:
> > The PyBullet tasks are periodic and the horizon is about 20 in a cycle. However, this periodic structure is not revealed to the learner and it is unknown whether the learner can capture this structure for more-efficient learning. Therefore, I tend to believe that from the learner’s view, the horizon is still 1000. Besides, as you said, the analysis is performed in a worst-case manner and thus may not explain the empirical observations well.
> >
> > Question 1:
> > Rajaraman et al. (2021) also analyzed the replay estimation technique with linear function approximation. In my understanding, Rajaraman et al. (2021) and this work use two different types of assumptions. Rajaraman et al. (2021) uses a uniform distribution assumption while this work uses a bounded density assumption. I cannot identify which one is weaker. Can the authors elaborate more on this point?

---

> > > ### Author Response · Authors · 2022-08-07
> > > **Re:**
> > >
> > > 1. As mentioned in the response to Reviewer hKQj, we note that although the environment is not as "easy" as one where the horizon length is revealed to the learner, the environment is significantly easier than a general task of 1000 episodes. A much harder example in this class could be one where 50 different tasks of horizon length 20 must be completed in sequence. Here, there is no significant information carryover from one task to the next and the sample complexity in practice is likely to be much higher compared to the case where the tasks are all identical.
> > >
> > > 2. The bounded density assumption is a relaxation of the uniform density assumption. In particular, if we set $c_{\min} = c_{\max}$ in our results, we recover the uniform distribution assumption condition in Rajaraman et al. (2021). We shall include this as a clarifying remark in the paper since it may not be apparent from the definitions. Also note that our proof technique is also based on significantly less specialized machinery than that used in Rajaraman et al. (2021), which heavily relies on the uniformity assumption. Thus it is the right stepping stone to extend these guarantees to even more general settings.

---

> > > > ### Comment · Reviewer_Jnmr · 2022-08-08
> > > > **Response to the authors**
> > > >
> > > > Thanks to the authors for the detailed response. Most of my concerns are well addressed. I increased my score from 5 to 7.

---

### Official Review · Reviewer_kTcU · 2022-07-14

**Rating:** 8
**Confidence:** 2
**Soundness:** 3 good
**Presentation:** 2 fair
**Contribution:** 4 excellent

**Summary:**


They propose a reply estimation way that uses a combination of simulated and empirical rollouts to optimally estimate expert moments. They show the finite sample guarantee that can enjoy both of the worlds of BC and moment matching.

**Questions:**

I wrote down the question above.

**Ethics Review Area:**

["I don’t know"]

**Limitations:**

Not applied.

**Strengths And Weaknesses:**

This paper's result is arguably strong. But at the same time, I feel some of the results are currently hard to understand. Many details are in the appendix currently, and hard to tell what are real assumptions and caveats from the main text.

(Strengths)

* Their proposed algorithm is sufficiently novel as far as I know.

* Their theoretical results are also very interesting. At least, I didn't expect we can enjoy both of the worlds of BC and moment matching in the function approximation setting.

(Weakness)
It looks like many important stuff are in the appendix. I can understand it might be due to the page constraint. The presentation would be much more improved by cutting some of the stuff.

* I personally feel Section 2 is too long? I guess it is introduced for readers to get some intuition. But, personally, I cannot get so much good intuition from these special instances, figure 2 and figure 3. (other reviewers might feel in a different way though) I would like to suggest authors shorten this section (just introduce BC/moment matching/replace estimation and what are known results in the tabular setting) without introducing these special instances.

* Some of the theorems are written in expectation. Some of the theorems are written in high probability.  This is confusing since comparisons would be not apple to apple sometimes.  Could you try to be consistent? If the authors cannot,  that’s fine. But, could you explain why?

* Some assumptions of theorems are hidden in the appendix. I think they should be in the main text as much as possible. (I think this can be done by shortening section 2 a lot)

	* For example, in Theorem 3.1, “instantiated with an appropriate discriminator F”… This should be in the man text. And It should be explained if it is not trivial stuff.
	* Theorem 2.  Did you assume the realizability of the reward function? But then, assumptions are stronger than BC? In this sense, you cannot directly compare with BC since the assumptions are different?  Does the same stuff apply to moment matching since moment matching might not require the reazaibility of the policy class?
	* Assumption 2 sounds strong. I am not sure you can really claim this result handles general function approximation. It looks like it only holds in the simple parametric case. How about more complete classes such as neural networks or random forests? (If not, “function approximation with parametric models” sounds more reasonable as a tile of the paragraph)
	* Theorem 3:“Appropriately instantiating —-“ This should be in the main text.
	* Line 321: “Whenever we are able to establish —- up to ‘’: How reasonable it is? When does it happen/

---

> ### Author Response · Authors · 2022-08-02
> **Re:**
>
> We thank the reviewer for their thoughtful questions and appreciate the opportunity to respond.
>
> Theorems in Expectation: The two results which are not high probability results are the expectation upper bound for moment matching in Theorem 3.1, and the constant probability lower bound for moment matching in Theorem 3.2. The former can easily be converted to an upper bound which holds in high probability essentially following the same proof. In particular, the only step that needs to be changed is the upper bound on the $L_1$ error for estimation of distributions with finite support, used in going from eq. (22) to (23), which is bounded in expectation in the current proof. A simple argument following the proof of Theorem 1 of [1] shows that with probability $\ge 1-\delta$, the $L_1$ distance between an empirical distribution and its population version scales as $\sqrt{S/N} \log(S/\delta)$. This enables the upper bound in Theorem 3.1 to be made into a high probability upper bound, showing that w.p. $\ge 1-\delta$, the imitation gap of MM is upper bounded by $H \sqrt{S/N} \log(HSN/\delta)$.
>
> Likewise, the lower bound in Theorem 3.2 can also be easily converted into a stronger high probability lower bound by changing the instance slightly to depend on delta. For lack of space, we skip the details, but using the anti-concentration of binomial distributions to bound the new probability of the new events $\mathcal{E}_1, \mathcal{E}_2$ and $\mathcal{E}_3$ (page 17 of supplementary material), one may arrive at the lower bound of $H * \sqrt{\log(1/\delta)/N}$ for fixed $\delta$ for the case of $S=2$.
> Thus, in summary, with a little effort the two results in the paper which are not high probability bounds can be converted into high probability bounds, with $polylog(1/\delta)$ dependence on the error probability.
>
> Appropriate Discriminator Class: Throughout the paper, we assume that $r \in \mathcal{F}$. We will make this more clear in the paper.
>
> Realizability of Reward Function: We assume that $r \in \mathcal{F}$ for convenience. Note that our algorithm, RE, admits the analogous “agnostic bound” in the non-realizable setting, where we bound the error against the best in-class performance. The results also extend to recover the appropriate agnostic bound when the policy class is not realizable and does not contain the expert policy. These results essentially follow from the fact that both BC and moment matching recover the appropriate agnostic bounds in the non-realizable setting. Moreover, note that BC can be thought of matching a particular kind of moment (see [2]) – and in that sense, our assumptions are no stronger.
>
> Assumption 2: We agree that “parametric function approximation” is a better terminology for this assumption and are happy to change this terminology in the paper.
>
> Line 321: We show conditions where parameter estimation matches the sample complexity of classification in the special case of linear classification in Appendix B.3. Note that linear classification already includes many expressive classes such as kernel methods. In general, the sample complexity of parameter estimation and classification are incomparable, and understanding the relationship between these quantities is a separate research thread of its own, which we do not go into detail in the paper.
>
> [1] Han, Y., Jiao, J., and Weissman, T. Minimax estimation of discrete distributions under L1 loss, 2015.
>
> [2] Swamy, G., Choudhury, S., Bagnell, J. A., & Wu, S. (2021, July). Of Moments and Matching: A Game-Theoretic Framework for Closing the Imitation Gap. In International Conference on Machine Learning (pp. 10022-10032). PMLR.

---

### Meta-Review · Area_Chair_AGUW · 2022-09-01

**Recommendation:** Accept
**Confidence:** Certain

**Metareview:**

Everyone in the review committee see this as a strong paper with a novel and practically useful replay estimation technique. Solid theoretical contribution with a meta theorem that can deal with general function approximation and weaker assumptions. Also strong empirical results on various benchmark. During rebuttal, authors were able to address most of the technical comments. Great work!

**Award:**

No

---

### Decision · Program_Chairs · 2022-09-14

Accept